# Straightforward Production Methods for Diverse Porous PEDOT:PSS Structures and Their Characterization

**DOI:** 10.3390/s24154919

**Published:** 2024-07-29

**Authors:** Rike Brendgen, Thomas Grethe, Anne Schwarz-Pfeiffer

**Affiliations:** 1Research Institute for Textile and Clothing (FTB), Niederrhein University of Applied Sciences, Webschulstr. 31, 41065 Moenchengladbach, Germany; 2Faculty of Textile and Clothing Technology, Niederrhein University of Applied Sciences, Webschulstr. 31, 41065 Moenchengladbach, Germanyanne.schwarz-pfeiffer@hs-niederrhein.de (A.S.-P.)

**Keywords:** smart textiles, porous conductive polymers, porous PEDOT:PSS, wearables, Production Methods for Smart Textiles

## Abstract

Porous conductive polymer structures, in particular Poly(3,4-ethylenedioxythiophene) polystyrene sulfonate (PEDOT:PSS) structures, are gaining in importance due to their versatile fields of application as sensors, hydrogels, or supercapacitors, to name just a few. Moreover, (porous) conducting polymers have become of interest for wearable and smart textile applications due to their biocompatibility, which enables applications with direct skin contact. Therefore, there is a huge need to investigate distinct, straightforward, and textile-compatible production methods for the fabrication of porous PEDOT:PSS structures. Here, we present novel and uncomplicated approaches to producing diverse porous PEDOT:PSS structures and characterize them thoroughly in terms of porosity, electrical resistance, and their overall appearance. Production methods comprise the incorporation of micro cellulose, the usage of a blowing agent, creating a sponge-like structure, and spraying onto a porous base substrate. This results in the fabrication of various porous structures, ranging from thin and slightly porous to thick and highly porous. Depending on the application, these structures can be modified and integrated into electronic components or wearables to serve as porous electrodes, sensors, or other functional devices.

## 1. Introduction

Conducting polymers (CPs), in the form of polyacetylene, were first discovered in 1977 [1]. CPs are functional polymers built from alternating single and double bonds, known as conjugated chain structure, which makes them essentially organic and yet electrically conductive. CPs have gained in importance since they can be used in various areas, including sensors [2,3], actuators [4,5], optoelectronics and solar cells [6,7] and electrocatalysts [8,9]. Furthermore, conducting polymers have garnered attention for wearable and skin-contact applications [10,11] owing to their biocompatibility.

Poly(3,4-ethylenedioxythiophene) polystyrene sulfonate, or PEDOT:PSS, is possibly the most widely used electrically conductive polymer. It is built of long chains of polystyrene sulfonate (PSS), on which shorter chains of polyethylenedioxythiophene (PEDOT) are attached by columbic attraction among them [12,13]. PEDOT is synthesized by the polymerization of the monomer 3,4-ethylenedioxytiophene (EDOT). Due to its rapid oxidation in air, PEDOT alone is, however, not stable. Therefore, a polyelectrolyte solution of PSS can be added to improve processability. When PEDOT is synthesized in the presence of PSS, an aqueous dispersion is obtained that can be processed into thin films. Thereby, PSS not only disperses and stabilizes PEDOT in water and other solvents but also functions as a counter-partner for primary doping [14,15]. PEDOT is a p-type doped polymer; thus, it is oxidized, and an electron is withdrawn from the polymer chain, leaving behind mobile holes that can migrate along the chain due to electron hopping [16].

The electrical conductivity of PEDOT:PSS increases significantly after treatment with secondary dopants such as acids, other organic compounds, or ionic liquids that produce conformational changes in the polymer chain or remove PSS. Secondary doping of PEDOT:PSS is often carried out with the organic solvent Dimethyl Sulfoxide (DMSO) [13,17,18,19,20,21]. It induces further changes to the polymer structure, which results in an increase in conductivity. The literature suggests that the critical dopant concentration of DMSO lies at about 0.57 wt%, however, to have a saturated conductivity, it is necessary to have a dopant concentration of about 4.0 wt%. PEDOT:PSS in aqueous solution has a concentration of 1–1.3 wt%, and the secondary dopant concentration should be higher or comparable to the amount of PEDOT:PSS to obtain highly conductive PEDOT:PSS films [19]. Another study came to similar conclusions and recorded the saturation of conductivity at 3.0 wt% [20]. Glycerol is the common name for propanetriol. It is a sugar alcohol and the simplest trivalent alcohol-triol. It is a viscous, hygroscopic liquid with water-binding and moisturizing properties. Besides these properties, glycerol can also be used for the conductivity enhancement of PEDOT:PSS [22,23]. The literature suggests an optimized concentration of glycerol at 0.24 g/mL [22].

Next to its high electrical conductivity, PEDOT:PSS is also convinced by its high transparency in visible light, high mechanical flexibility, high thermal stability, and good oxidation resistance [14]. It forms gel-like particles of 50 nm diameter that are dispersed in water or other solvents. The dispersion can be applied by conventional printing and coating methods, such as spin-coating, slot-die coating, or screen printing [12].

(Porous) conductive polymer structures are becoming increasingly significant, since their employability as (electrochemical) sensors [24,25], hydrogels [26,27] or supercapacitors [28,29]. Basically, production methods of porous conductive structures can be classified into template-free (direct synthesis) and template methods, depending on whether supporting materials (templates) are used or not. Template-free production methods include, but are not limited to, direct electro-polymerization, in situ chemical polymerization, electrospinning, and spin coating. Template-free methods have the advantage of simplifying the processes, though the morphologies of the porous structures are difficult to control. Template methods, on the other hand, usually comprise several production steps that include: preparation/assembly of the template structure; coating or infiltration of the template gaps with polymer material; and removal of the templates, though in some instances removal is not required [24].

PEDOT:PSS specifically is often prepared as a porous structure in various production methods, comprising the usage of diverse scaffolding or filling materials [30,31,32,33] as well as ice-templating [34,35,36] or other methods. Porous PEDOT:PSS structures are prominently used in various applications due to their unique properties such as high conductivity, biocompatibility, environmental stability, and porosity. They are utilized in tissue engineering for providing scaffolds that support cell proliferation and differentiation, in drug delivery systems for controlled release rates of encapsulated drugs, and in bioelectronics due to their high electrical conductivity and stability in aqueous environments. Additionally, they are applied in environmental pollution control for the adsorption and removal of pollutants, in energy storage devices like supercapacitors and batteries to enhance surface area and charge storage capacity, and in sensors and actuators for their sensitivity and conductive properties [37,38]. Specific sensor applications of porous PEDOT:PSS include:**Humidity Sensors:** Leveraging the hygroscopic nature of PEDOT:PSS, humidity sensors can detect changes in moisture levels through variations in electrical resistance [37]. For instance, flexible humidity sensors using PEDOT:PSS nanowires show high sensitivity (5.46%) and ultrafast response times (0.63 s), suitable for applications like human breath testing [39].**Pressure Sensors:** PEDOT:PSS can be embedded into compressible structures to create highly sensitive pressure sensors. One example is a pressure sensor made by coating a thin layer of PEDOT:PSS on a pyramidal PDMS substrate, enabling precise pressure detection, such as monitoring blood pulses at the wrist [40]. Other methods include using sponge-like substrates made from materials such as PU [30] or melamine [41]. In these designs, the PEDOT:PSS-modified foam is integrated into a shoe sole for real-time body weight distribution analysis and a detailed representation of walking dynamics [30] or attached to different body parts for various motion detection applications [41].**Strain Sensors:** PEDOT:PSS strain sensors are used in flexible electronics and health monitoring. While intrinsic PEDOT:PSS is brittle and prone to breakage, making it unsuitable for stretchable strain sensors, its stretchability can be significantly enhanced. This is achieved by blending PEDOT:PSS with elastomers or by incorporating it into textile fibers or aerogels, thereby improving its flexibility and durability [37]. A notable application involves a strain sensor combining PEDOT:PSS with a biocompatible polymer, achieving a conductive elastomer with a maximum strain of 230%, ideal for flexible and wearable electronics [42]. Additionally, PEDOT:PSS textile fibers can construct wearable sensors due to their excellent flexibility and skin adhesion [43].**Temperature Sensors:** PEDOT:PSS is a promising material for wearable temperature sensors due to its flexibility and high thermal response. Its sensitivity arises from temperature-induced microstructural changes in the hygroscopic PSS-rich shell [37]. Enhancements in thermal sensitivity have been achieved through secondary doping and thermal expansion techniques. For instance, microcracks were created in a PDMS substrate, resulting in a temperature sensitivity of 4.2%/°C for a wearable temperature sensor [44].**Electrochemical Sensors:** Electrochemical sensing, leveraging porous conducting polymers, offers significant advantages in terms of sensitivity and response time. Due to their large specific surface area and high electrical conductivity, porous conducting polymers provide abundant active sites and reduce charge-transfer resistance. These attributes enhance the interaction between the target molecules and the sensor surface, leading to improved detection capabilities [24]. For instance, PEDOT was doped with a pure insoluble ionic liquid, resulting in a nanoporous, highly conductive, and stable microstructure that demonstrated excellent electrocatalytic activity for the oxidation of dopamine [45].**Gas Sensors:** Porous conductive polymer structures offer significant advantages in gas sensors due to their high surface area, which enhances gas adsorption and interaction. This increased surface area allows for more effective interaction between the conductive polymer and the target gas, improving the sensor’s sensitivity and response time. Additionally, the porous structure facilitates the diffusion of gases throughout the material, leading to more uniform detection and a reduced response time. The flexibility and lightweight nature of conductive polymers further enhance their suitability for wearable and portable gas sensors. For instance, a PEDOT and reduced graphene oxide nanocomposite were developed for room-temperature NH_3_ gas detection [46].

However, the production methods for porous conductive polymers are often complex, time-consuming, or require a large amount of chemicals, which is especially disadvantageous when being used in wearables or smart textiles. In this study, we present straightforward methods derived from textile technologies for producing diverse porous PEDOT:PSS structures that can be used as electrode materials [47,48] or sensors [37,49]. The employed production methods include the incorporation of micro cellulose, using a blowing agent, creating a sponge-like structure, and spraying onto a porous base substrate. The resulting samples are evaluated manually for mechanical stability and assessed in terms of optical properties, porosity, and electrical resistance.

## 2. Materials and Methods

The following sections describe, firstly, the different production methods for porous PEDOT:PSS structures and, secondly, the employed characterization methods.

### 2.1. Production of Porous PEDOT:PSS Structures

For the production of porous PEDOT:PSS structures, different methods are tested, which include the use of blowing agents, the incorporation of scaffolding material, and the spraying of a porous base substrate. For that, the common ingredient is PEDOT:PSS Clevios PH1000, purchased from Heraeus Deutschland GmbH + Co. KG (Leverkusen). For all trials, if not stated otherwise, PEDOT:PSS is doped with DMSO (99.7%, Sigma Alderich/Merck KGaA, Darmstadt, Germany) and glycerol (≥98%, Carl Roth GmbH + Co.KG, Karlsruhe, Germany). Therefore, PEDOT:PSS is mixed with 3.0 wt% DMSO and 0.24 g/mL glycerol and mechanically stirred for 20 min.

#### 2.1.1. PVA/Micro Cellulose/PEDOT:PSS Networks

The production of PVA/micro cellulose/PEDOT:PSS networks starts with the preparation of a 8.0 wt% PVA (Sigma Aldrich/Merck KGaA, Darmstadt, Germany) solution. Subsequently, 2.0 g DMSO, 0.1 g of 12.5 wt% glutaraldehyde (GA) from Carl Roth GmbH + Co. KG (Karlsruhe, Germany), and a specified quantity of PEDOT:PSS (10, 20, 30%) are introduced to the solution. The mixture is stirred in the high-speed disperser Dispermat of VMA-Getzmann GmbH (Reichshof, Germany) at 2000 rpm for 10 min. During mixing, 0.1 g of 10 wt% hydrochloric acid (HCl) from Carl Roth GmbH + Co. KG (Karlsruhe, Germany) is added to the solution. Finally, the solution undergoes degassing through ultrasonic treatment in the ultrasonic bath Sonorex Digitec of Bandelin electronic GmbH + Co. KG (Berlin, Germany) for an additional 10 min. This production process is based on the hydrogel preparation of Y.-F. Zhang et al. (2020) [50]. In a subsequent step, 20 μm of micro cellulose from Sigma Aldrich/Merck KGaA (Darmstadt, Germany) is added in 0, 2, 4, 8, and 16% to the solution. Micro- or nanocellulose has been added to PEDOT:PSS before to achieve highly conductive porous cellulose/PEDOT:PSS nanocomposite paper [32]. Resulting in the following samples (Table 1):

The solutions are squeegeed onto a Teflon foil with a wet film thickness of 500 µm and left to dry at 25 °C for 24 h.

#### 2.1.2. Porous PU/PEDOT:PSS Sponge

A porous PU/PEDOT:PSS sponge is prepared by initially incorporating sugar particles into a polymer matrix and subsequently dissolving them after curing. The manufacturing technique is based on [33,51] that both use the silicone-based material Ecoflex 00-30 as sponge material. In this research, however, commercially available PU-based coating dispersion Tubicoat PU60 from CHT Germany GmbH (Tübingen, Germany) is used instead of the hydrophobic silicone. The PU dispersion is mixed with doped PEDOT:PSS in a ratio of 1:2. Tubicoat thickener LP is added at 2.2% to the dispersion before mixing in the disperser. Sugar is added in varying ratios (1:1, 1:1.5, 1:2, 1:2.5) to the dispersion and mixed by hand before directly drying the casted material at 97 °C for 2 h. Resulting in the following samples (Table 2):

#### 2.1.3. Puffed PEDOT:PSS Structures

Puffy printing paste is used in textile printing to achieve special printing effects like gloss, glitter, pearl, and metal optics. Also, it has been used in smart textile applications before, but as an insulation layer [52,53]. The commercially available puffy printing paste Tubiscreen EX-TS-FF (CHT Germany GmbH, Tübingen, Germany) is mixed with doped PEDOT:PSS in different ratios; see Table 3. The dispersion is squeegeed onto a Teflon foil with a wet film thickness of 500 µm and subsequently dried at 160 °C. Upon higher addition of PEDOT:PSS content, the drying time increases up to approximately 7 min until the samples are fully hardened.

#### 2.1.4. Starch-Based Puffed PEDOT:PSS Structures

Puffy printing paste can also be made by mixing flour, baking soda, water, and a small amount of salt. Instead of adding water, the doped PEDOT:PSS solution can be added. In that way, flour and doped PEDOT:PSS are mixed in a ratio of 1:1.5, and baking soda is added to that mixture in different quantities (7.5, 15, and 30%), resulting in the following samples (Table 4):

The resulting paste is cast into a mold and microwaved at 800 W for 10 s.

#### 2.1.5. Sprayed PEDOT:PSS Non-Wovens

Porous non-wovens are sprayed with doped PEDOT:PSS. Therefore, a spaying nozzle with a diameter of 0.8 mm is used, and a spraying distance of 10 cm as well as a pressure of 2.0 bar are set. A single spraying layer is applied for 5 s before drying at 100 °C for 3 min. Samples with different numbers of spraying layers are prepared; see Table 5:

### 2.2. Characterization Methods

The comprehensive characterization of the porous PEDOT structures employs a multifaceted approach, incorporating visual and tactile assessments, microscopic imaging, porosity measurements, and electrical resistance evaluations. Detailed methodologies are elaborated upon below.

#### 2.2.1. Visual and Haptic Assessment

The samples undergo rigorous visual and tactile evaluation. Manual bending and stretching procedures are applied to meticulously assess the flexibility and structural integrity of the samples, with particular emphasis on their bendability and overall robustness.

#### 2.2.2. Microscopic Imaging

Microscopic imaging was performed utilizing the TM4000 Plus scanning electron microscope (SEM) from Hitachi High-Tech Europe GmbH (Krefeld, Germany). Both longitudinal and cross-sectional images were acquired at multiple magnifications to facilitate a thorough and detailed analysis. Cross-sectional imaging included meticulous thickness measurements to ensure accuracy. Furthermore, elemental analysis was conducted using the energy dispersive X-ray spectroscopy (EDS) unit from Bruker Corporation (Billerica, MA, USA), precisely at 200× magnification, to deliver a detailed elemental composition at a specific measurement point.

#### 2.2.3. Porosity Measurement

Porosity, with a specific focus on pore size, was characterized using the Topas PSM 165 porosity measurement device, which is specifically designed for evaluating porous materials such as non-wovens and woven fabrics. This device operates based on the principle of capillary liquid porometry. For the porosity measurements, samples were precisely cut into 1 cm × 1 cm sections and placed into the specimen holder of the device. The green specimen holder, featuring a surface diameter of 0.28 cm^2^ (the smallest opening), was utilized. Each substrate underwent three separate tests at different locations, and the mean value was calculated to ensure accurate and reliable results.

#### 2.2.4. Electrical Resistance Measurement

For electrical resistance measurements, the bench-type digital multimeter 4000 from PeakTech Prüf- und Messtechnik GmbH (Ahrensburg, Germany) was employed. Crocodile clamps with measurement tips were utilized for contacting, and a measuring width of 1 cm was defined, as only small samples were produced. Each sample was tested five times at five different locations, and the mean value was calculated to ensure precision and reliability in the results (Figure 1).

## 3. Results

This section presents the results of the characterization methods applied to each individual production method.

### 3.1. PVA/Micro Cellulose/PEDOT:PSS Networks

The visual and haptic assessment indicates that samples without micro cellulose resemble a thin, transparent film. The addition of micro cellulose imparts a paper-like texture and enhances structural integrity. However, an excessive micro cellulose content (from 8% onwards) results in brittleness, causing the material to break easily upon bending. Figure 2A demonstrates the bendability of a sample with 20% PEDOT:PSS and 2% micro cellulose. In contrast, a sample with 16% PEDOT:PSS (Figure 2B) lacks flexibility and breaks when folded.

In a subsequent step, the samples are characterized in the SEM. It has to be noted that it is not possible to take microscopic pictures of the samples containing no micro cellulose (just cross-sectional wise), as their film-like surface is too smooth, shine-through, and thin to be detected. Figure 3 shows exemplary SEM images of the samples containing 20% PEDOT:PSS and different micro cellulose contents (2–16%). Especially when comparing Figure 3A (2% micro cellulose) to Figure 3D (16% micro cellulose), the density of micro cellulose fibers increases. However, even a small amount of micro cellulose (2%) results in complete coverage with fibers.

Figure 4 displays the cross-sectional view of exemplary samples from the P10-30C0-16 series, with (B, C) and without (A) micro cellulose. The sample without micro cellulose has a mean layer thickness of 25.9 µm, while the addition of micro cellulose proportionally increases the layer thickness. The following layer thicknesses were obtained for samples with different micro cellulose concentrations (Table 6):

EDS analysis at 200× magnification focuses on the element sulfur (S) for PEDOT:PSS detection, as the organic materials (micro cellulose, PVA, PEDOT:PSS) contain carbon (C), nitrogen (N), and oxygen (O), which are not indicative of PEDOT:PSS. Sulfur is detectable in both PEDOT and PSS. Figure 5 shows S-mapping for samples with varying PEDOT:PSS content (P10C4, P20C4, and P30C4), indicating a relatively uniform distribution of S across all samples, with no significant increase in mapping points with higher PEDOT:PSS concentrations.

Quantitative EDS analysis of atom percentages reveals no significant change in the atom percentage of S (Figure 5). The mean values of S for samples with 10%, 20%, and 30% PEDOT:PSS are all between 0.64% and 0.69% (Figure 6).

Porosity measurements on the PVA/micro cellulose/PEDOT:PSS network yield no results, even with larger specimen holders. This suggests that the pore size may fall outside the device’s measurement range (0.25–130 µm) or that the samples may lack porosity.

Figure 7A–E shows resistance measurements of the PEDOT:PSS structures combined with PVA and micro cellulose. Each sample undergoes five measurements at five locations, and mean values as well as the standard deviation are presented. Figure 7A provides an overview, while Figure 7B–E offer detailed views. The resistance decreases significantly with increasing PEDOT:PSS content, dropping from ~1320 kΩ (P10C0) to ~80 kΩ (P30C0). The most substantial reduction occurs between 10% (P10C0) and 20% PEDOT:PSS (P20C0), with a decrease of ~1138 kΩ. Additional micro cellulose up to 4% further reduces resistance, but beyond this content, resistance increases. The lowest resistance (9.68 kΩ) is observed in the sample with 30% PEDOT:PSS and 4% micro cellulose (P30C4). The standard deviation is proportional to the electrical resistance and decreases significantly with lower electrical resistance.

### 3.2. Porous PU/PEDOT:PSS Sponge

Figure 8 shows the produced PU/PEDOT:PSS sponges and their bendability. The sponges are stiff, brittle, and significantly thicker than other porous structures, with no noticeable differences in touch and bendability between samples with different PU/PEDOT:PSS to sugar ratios.

Figure 9 presents SEM images at 100× magnification, illustrating samples with varying sugar amounts. Increased sugar content enlarges the holes in the structure, as seen in Figure 9A,B (PUP1S1-1.5) compared to Figure 9C,D (PUP1S2-2.5).

Figure 10 shows a cross-section of sample PUP1S2 at 50× magnification, revealing a mean thickness of 1.3 mm and visible channels or cavities.

EDS analysis at 200× magnification focuses on sulfur (S), characteristic of PEDOT:PSS. Figure 11 shows EDS mapping for PUP1S1 (A) and PUP1S2 (B), with no variation in the number of mapping points, indicating constant PEDOT:PSS concentration.

Compared to other porous PEDOT:PSS samples, S-mapping points are fewer (Figure 11). Atomic percentage data in Figure 12 show that sulfur is nearly non-detectable, averaging only 0.11 atom-%.

Figure 13 presents pore size measurements of PU/PEDOT:PSS sponges. Increasing sugar content enlarges the mean pore size from 111.9 µm (PUP1S1) to 174.0 µm (PUP1S2.5). Thereby, especially the smallest pores grow from 11.9 µm to 56.0 µm.

Figure 14 shows electrical resistance measurements, with no clear variation among samples produced with different sugar contents, averaging around 63.78 kΩ. This consistency indicates that porosity does not affect electrical resistance.

### 3.3. Puffed PEDOT:PSS Structures

Upon examination of the PEDOT:PSS samples subjected to the puffing process, a notable inverse correlation between the concentration of PEDOT:PSS and the degree of surface wrinkling is observed, as evidenced by the images presented in Figure 15. Concomitantly, an increase in PEDOT:PSS content leads to a progressive deterioration of the structural integrity, resulting in enhanced pliability but diminished dimensional stability of the specimens. Notwithstanding these alterations in physical properties, all samples retain a high degree of flexural compliance and malleability, as demonstrated visually in Figure 15.

Figure 16 presents representative scanning electron micrographs at 200× magnification, elucidating the morphological characteristics of the puffed PEDOT:PSS structures. The micrographs, labeled A through F, correspond to incrementally increasing PEDOT:PSS concentrations ranging from 0% to 400%. These micrographs reveal a notable presence of spherical voids, which are attributed to the evolution of gaseous phases during the drying process, a phenomenon induced by the incorporation of Tubiscreen EX-TS-FF microbeads. A discernible inverse relationship is observed between the PEDOT:PSS concentration and the prevalence of these void structures. This trend is particularly evident when contrasting Figure 16A (0% PEDOT:PSS, designated as TS1P0) with Figure 16F (400% PEDOT:PSS, designated as TS1P4). Concurrently, a progressive enhancement in the overall electron emission intensity is noted with increasing PEDOT:PSS content. This augmented brightness is consistent with the established principle that materials exhibiting higher electrical conductivity manifest increased electron emission in scanning electron microscopy, as supported by previous studies [54] and visually apparent in the sequence of micrographs presented in Figure 16.

Figure 17 presents a representative cross-sectional scanning electron micrograph of the TS1P4 sample at 100× magnification. The micrograph reveals a complex internal structure characterized by the presence of numerous cavities. These void spaces are hypothesized to be the result of the rupture and subsequent collapse of the incorporated microbeads during the manufacturing process.

Quantitative analysis of the cross-sectional image yields an average thickness of 301.5 µm for the TS1P4 sample. However, it is imperative to note that the precise demarcation of the sample boundaries is confounded by the inherent surface irregularities. These topographical variations introduce a degree of uncertainty in the thickness measurements, necessitating a cautious interpretation of the dimensional data.

The observed microstructure, with its network of interconnected cavities, has significant implications for the material’s properties, including its porosity, specific surface area, and potentially its electro(-chemical) performance. These structural features may play a crucial role in the functionality of the material as a component in textile-based electro(-chemical) devices or sensors.

In conjunction with scanning electron microscopy (SEM) imaging, energy-dispersive X-ray spectroscopy (EDS) was employed for elemental mapping, with particular emphasis on the spatial distribution of sulfur, a key constituent of PEDOT:PSS. Figure 18A–C presents a series of elemental maps that elucidate the correlation between PEDOT:PSS concentration and sulfur abundance.

A pronounced gradient in sulfur concentration is observed across the samples with varying PEDOT:PSS content. In Figure 18A, corresponding to sample TS1P0 (0% PEDOT:PSS), the sulfur map exhibits minimal signal intensity, with the sparse detection points primarily attributed to background noise inherent to the SEM-EDS system. In stark contrast, Figure 18C, representing sample TS1P4 (400% PEDOT:PSS), reveals a markedly elevated sulfur signal. This is visually represented by a dense array of detection points, culminating in a continuous orange-hued map that signifies a high sulfur concentration.

The progressive intensification of the sulfur signal from Figure 18A–C provides compelling evidence for the successful incorporation and homogeneous distribution of PEDOT:PSS within the material matrix. This elemental mapping technique offers valuable insights into the spatial distribution of the conductive polymer, which is crucial for understanding and optimizing the electro(-chemical) properties of these materials.

This elemental analysis validates the previously observed trends in SEM imaging and provides a quantitative basis for assessing the PEDOT:PSS integration. The correlation between sulfur distribution and PEDOT:PSS concentration underscores the efficacy of the fabrication process in achieving controlled polymer loading, a critical factor in tailoring the electrical and electrochemical properties of these textile-based electrode materials.

Quantitative analysis of the elemental composition, as depicted in Figure 19, verifies the qualitative observations from the EDS mapping. As anticipated, the predominant atomic species are carbon (C), nitrogen (N), and oxygen (O), which constitute the primary structural components of the material matrix. Of particular interest in this study is the atomic percentage of sulfur, which serves as a diagnostic marker for the presence and concentration of PEDOT:PSS.

A clear positive correlation is observed between the PEDOT:PSS content and the atomic percentage of sulfur. In the control sample devoid of PEDOT:PSS (TS1P0), the sulfur content is negligible (0.09 atom-%), likely attributable to background noise or trace contaminants. In contrast, the sample with the highest PEDOT:PSS loading (TS1P4) exhibits a sulfur content of 0.34 atom-%, representing a nearly four-fold increase.

The relationship between PEDOT:PSS concentration and sulfur atomic percentage demonstrates a biphasic trend. Up to a PEDOT:PSS concentration of 200%, the atomic percentage of sulfur increases in an approximately linear fashion. However, beyond this threshold, the rate of increase in sulfur content diminishes, suggesting a potential saturation effect or a change in the incorporation mechanism of PEDOT:PSS at higher concentrations.

This non-linear behavior at elevated PEDOT:PSS concentrations warrants further investigation. It may indicate the onset of polymer aggregation, changes in the material’s microstructure, or limitations in the uniform dispersion of PEDOT:PSS within the matrix. Such phenomena could have significant implications for the electrical conductivity, electrochemical activity, and overall performance of these materials in sensing applications.

The quantitative elemental analysis provides crucial insights into the compositional changes induced by varying PEDOT:PSS concentrations. This data, in conjunction with the morphological observations from SEM imaging, forms a comprehensive basis for understanding the structure-property relationships in these textile-based electrode materials. The observed trends in sulfur content offer valuable guidance for optimizing PEDOT:PSS loading to achieve the desired electrical and electrochemical characteristics, depending on the application.

Figure 20 presents the results of the porosity analysis conducted on the puffed PEDOT:PSS structures. For comparative purposes, a control sample containing only Tubiscreen EX-TS-FF without PEDOT:PSS (TS1P0) was included in the analysis.

The data reveal a distinct correlation between PEDOT:PSS concentration and pore size distribution. A monotonic increase in the minimum pore diameter is observed with increasing PEDOT:PSS content. Specifically, the minimum pore size ranges from 5.1 µm in the control sample (TS1P0) to 44.7 µm in the sample with the highest PEDOT:PSS loading (TS1P4), representing an approximately nine-fold increase.

Similarly, the mean pore diameter exhibits a positive correlation with PEDOT:PSS concentration. The average pore size increases from 19.0 µm in the control sample (TS1P0) to 52.1 µm in the sample with 400% PEDOT:PSS (TS1P4). Notably, the most significant increment in mean pore size is observed between the control sample (TS1P0, 19.0 µm) and the sample with 100% PEDOT:PSS (TS1P1, 38.0 µm), indicating a possible threshold effect in the influence of PEDOT:PSS on pore formation.

It is worth noting that while clear trends are observed for the minimum and mean pore sizes, the data for the maximum pore size and the most prevalent pore size (highest share) do not exhibit conclusive patterns across the PEDOT:PSS concentration range.

The observed changes in pore size distribution with increasing PEDOT:PSS content have significant implications for the material’s properties and potential applications. The enlargement of pores could lead to increased surface area and enhanced ion accessibility, which are crucial factors in electro(-chemical) sensing applications. However, it may also affect the mechanical properties and stability of the material.

These findings provide valuable insights into the microstructural evolution of the puffed PEDOT:PSS structures as a function of polymer loading. The relationship between PEDOT:PSS concentration and pore size distribution is a critical consideration in optimizing these materials for use in textile-based electrode and sensing applications. Further investigation into the mechanisms underlying this pore size modulation and its impact on electro(-chemical) performance is warranted to fully leverage these structural changes in sensor design.

Figure 21 illustrates the resistance measurements of the porous PEDOT:PSS structures fabricated by incorporating varying amounts of doped PEDOT:PSS into the puffy printing paste Tubiscreen EX-TS-FF. The data reveals an exponential decrease in resistance as the PEDOT:PSS content is linearly increased. The electrical resistance drops markedly from 57.6 kΩ at 100% PEDOT:PSS (TS1P1) to 3.66 kΩ when the PEDOT:PSS content is doubled to 200% (TS1P2). Further increasing the PEDOT:PSS content continues to reduce the electrical resistance, albeit at a diminished rate compared to the initial decrease. At 400% PEDOT:PSS (TS1P4), the resistance reaches a low value of 0.9 kΩ. These findings suggest a percolation threshold is achieved between 100% and 200% PEDOT:PSS loading, beyond which additional PEDOT:PSS has a lessened impact on enhancing conductivity, indicating the system approaches saturation. The standard deviation also decreases significantly with lower electrical resistance, making the error bars not visible from TS1P2 onwards due to the scaling of the figure.

### 3.4. Starch Based Puffed PEDOT:PSS Structures

Figure 22 shows that starch based puffed PEDOT:PSS samples exhibit flexibility and bendability, whereby the amount of baking soda does not influence these properties, though it increases height and the blowing up effect. The samples are soft and doughy and therefore susceptible to water.

Microscopic imaging reveals that increasing the amount of baking soda not only amplifies the blowing-up effect but also leads to an augmentation in both the quantity and dimensions of the pores within the material. Figure 23A,B provide a top-down perspective of the substrates, indicating that the pore size undergoes a more pronounced increase compared to the pore density. This observation is further corroborated by the cross-sectional views presented in Figure 23C,D, which offer a complementary visual confirmation of the pore size enhancement. The cross-sectional images clearly showcase the internal structure of the samples, providing compelling evidence for the direct relationship between baking soda content and pore size expansion. These microscopic investigations offer valuable insights into the morphological evolution of the material as a function of baking soda concentration, highlighting the tunability of pore characteristics through judicious control of the blowing agent.

Cross-sectional layer thickness measurements, as illustrated in Figure 24, reveal a substantial increase in layer thickness from 1.94 mm for sample F1P1.5BS1 to 2.85 mm for sample F1P1.5BS2. This marked increase in thickness can be attributed to the higher baking soda content in the F1P1.5BS2 sample, which enhances the blowing-up effect during the fabrication process. It is noteworthy that the layer thickness measurement for sample F1P1.5BS3 could not be accurately determined due to the limitations imposed by the excessively high magnification of the scanning electron microscope (SEM). This constraint highlights the challenges associated with characterizing the thickest samples within the experimental set. Nevertheless, it can be confidently stated that the samples in this series, particularly F1P1.5BS3, exhibit the greatest layer thicknesses among all the porous PEDOT:PSS structures produced in this study. The exceptional thickness of these samples underscores the effectiveness of the baking soda-based blowing agent in creating highly expanded and porous PEDOT:PSS structures, which hold promise for applications demanding enhanced surface area and material volume.

The samples were further investigated using energy-dispersive X-ray spectroscopy (EDS) mapping and analysis, as presented in Figure 25 and Figure 26. The EDS mapping results indicate that increasing the baking soda content does not lead to a significant reduction in the number of mapping points across the different samples. This observation suggests that the overall elemental composition remains relatively consistent, despite variations in the baking soda concentration. However, it is important to acknowledge the inherent challenges associated with detecting elements in highly porous structures. The presence of large voids and intricate pore networks can hinder the accurate acquisition of EDS signals, as electrons may penetrate deeper into the sample without generating characteristic X-rays from the surface. This phenomenon is particularly evident in Figure 25C, which corresponds to sample F1P1.5BS3. The EDS mapping of this sample reveals a notably lower density of mapping points within the large pore region. The reduced signal intensity can be attributed to the increased porosity and the consequent limitation in collecting representative X-ray emissions from the material. Despite these challenges, the EDS analysis provides valuable insights into the elemental distribution within the porous PEDOT:PSS structures and highlights the need for careful consideration of the impact of porosity on the acquisition and interpretation of EDS-data.

The quantitative analysis (Figure 26) shows the elementary composition of the starch-based puffed PEDOT:PSS structures. Elements such as sodium (Na) and phosphor (P) are visible compared to the other PEDOT:PSS structures due to the presence of baking soda. Therefore, their atomic percentage increases with rising baking soda content. Chlorine (Cl) is visible due to the presence of chloride in the salt (NaCl) that was added to the mixture. Sulfur (S) as the characteristic element for PEDOT:PSS is little detectable, and its atomic percentage decreases from 0.32 atom-% for F1P1.5BS1 to 0.20 atom-% for F1P1.5BS3.

During porosity measurements (Figure 27), the increase in pore sizes could be confirmed. In that way, the mean pore size increases from 18.5 µm (F1P1.5BS1) to 56.0 µm (F1P1.5BS3). Notably, there is a discernible increase in pore size, particularly for the largest pores identified (F1P1.5BS1: 41.9 µm, F1P1.5BS3: 108.7 µm).

When measuring electrical resistance (Figure 28), all samples, regardless of their baking soda content, register at approximately 420 to 440 kΩ due to their constant PEDOT:PSS content. The standard deviation also remains relatively constant.

### 3.5. Sprayed PEDOT:PSS Non-Wovens

The intensity of the blue coloration in the PEDOT:PSS-sprayed samples exhibits a direct correlation with the number of spraying layers applied. The application of two layers of PEDOT:PSS results in a light blue tint, while increasing the number of spraying layers to 16 yields a deep, dark blue hue. This progressive intensification of the blue coloration can be attributed to the increased concentration of PEDOT:PSS on the surface of the non-woven substrate as more layers are deposited.

Interestingly, the mechanical properties of the sprayed samples also appear to be influenced by the number of spraying layers. Samples with fewer spraying layers (NWP2, NWP4) exhibit a brittle and stiffer texture, likely due to the limited amount of PEDOT:PSS present, which may not be sufficient to form a continuous, flexible coating on the non-woven fibers. In contrast, as the number of spraying layers increases, the non-woven substrate becomes progressively softer. This softening effect can be attributed to the increasing presence of PEDOT:PSS, which forms a more cohesive and flexible matrix around the non-woven fibers.

It is worth noting that the samples with the highest number of spraying layers (NWP16) exhibit a unique tactile property, almost as if they have not completely dried. This characteristic can be attributed to the presence of glycerol in the PEDOT:PSS formulation. Glycerol, a well-known humidifying agent, has the ability to retain moisture and prevent complete drying of the PEDOT:PSS coating. Despite this apparent moisture retention, all the samples maintain their bendability, although they do not exhibit high stretchability (Figure 29).

These findings underscore the significant influence of the number of PEDOT:PSS spraying layers on both the visual appearance and mechanical properties of the non-woven substrates. The ability to control the coloration intensity and modulate the texture of the samples by varying the number of spraying layers offers a facile and effective means of tailoring the properties of these conductive non-woven composites for specific applications. Further investigation into the effects of glycerol content and drying conditions on the mechanical and electrical properties of these samples could provide valuable insights into optimizing their performance and stability for various technological applications.

Figure 30 displays SEM images (100× magnification) of the PEDOT:PSS-sprayed non-wovens. It is noticeable that with increasing layers of PEDOT:PSS, the surface of the non-woven is wetted, and the individual filaments become glued together. This is especially evident when comparing Figure 30A,E.

Figure 31 shows the exemplary cross-section of the sample NWP8 with thickness measurements at 200× magnification. From all examined samples, the PEDOT:PSS sprayed samples are thinnest (except PVA/micro cellulose/PEDOT:PSS networks, but not porous), and the one depicted has a mean thickness of 97.1 µm. In contrast to other porous samples, no obvious channels or cavities are visible that permeate the material.

In addition to the microscopic examination, EDS analysis has been conducted. Once again, special attention is given to the element sulfur, which is characteristic of the presence of PEDOT:PSS. Figure 32 exemplifies the EDS mapping of the samples NWP0 (A), NWP4 (B), and NWP16 (C). The distinct increase in mapping points for the element sulfur serves as an indicator of the presence of PEDOT:PSS.

This can also be demonstrated quantitatively by examining the atomic percentage in Figure 33. The atomic percentage of S increases with the number of sprayed layers. S is nearly non-detectable in sample NWP0, and the small amount present is attributed to background noise from the instrument. For sample NWP16, the atom-% of S is 1.44, the highest percentage detected during this measurement series for all porous PEDOT:PSS samples. Additionally, the increase in atomic percentage from sample to sample is nearly linear, approximately doubling as the number of sprayed layers doubles.

The porosity of doped PEDOT:PSS-sprayed non-wovens was quantitatively assessed via porosity measurements (Figure 34). The mean pore size exhibits a distinct inverse correlation with the number of sprayed layers. NWP0, the non-woven substrate prior to PEDOT:PSS application, demonstrates a mean pore size of 23.0 µm. Subsequent application of 16 PEDOT:PSS layers via spray deposition results in a reduction in the mean pore size to 16.0 µm. Microscopic examination suggests that the spray deposition process leads to wetting of the non-woven surface, with PEDOT:PSS filling the interstitial spaces between individual filaments, thereby decreasing pore size. The data indicate that this pore size reduction is particularly pronounced for larger pores, as evidenced by the decrease in maximum pore size as a function of spray cycle number. The most substantial reduction in maximum pore size is observed for sample NWP2 (30.8 µm) relative to the uncoated substrate NWP0 (40.5 µm). These findings elucidate the morphological evolution of the non-woven microstructure upon PEDOT:PSS deposition and have important implications for tailoring porosity in conductive non-woven materials.

Figure 35 unambiguously demonstrates the inverse relationship between the number of PEDOT:PSS spray deposition cycles and the electrical resistance of the resultant non-woven composite. The application of a mere two layers of PEDOT:PSS yields a substantial resistance of 38.15 kΩ. Doubling the number of deposited layers to four results in a precipitous decrease in electrical resistance to 409 Ω, a reduction in nearly two orders of magnitude. Further increasing the number of spray deposition cycles to 16 leads to an ultimate resistance of 31.34 Ω, representing an additional order of magnitude decrease. The most pronounced reduction in electrical resistance is observed between samples NWP2 and NWP4, suggesting the formation of a percolated conductive network on the non-woven substrate at this critical PEDOT:PSS loading threshold. Additionally, there is a noticeable reduction in standard deviation between NWP2 and NWP4, such that the standard deviation becomes unrecognizable from NWP4 onwards due to the figure’s scaling. These findings highlight the efficacy of spray deposition as a means of controlling the electrical properties of PEDOT:PSS-based non-woven composites and provide valuable insight into the percolation behavior of these materials. The ability to tune resistance over several orders of magnitude through facile spray processing holds great promise for the rational design and optimization of conductive non-woven materials for a wide range of applications.

## 4. Discussion

The PVA/micro cellulose/PEDOT:PSS networks (P10-30C0-16) exhibit a spectrum of characteristics, ranging from film-like to paper-like, reliant upon their micro cellulose content. Samples lacking micro cellulose (P10C0, P20C0, and P30C0) form exceptionally thin, self-standing, film-like substrates. The incorporation of micro cellulose induces a slight increase in thickness. Moreover, the mechanical properties are controlled by micro cellulose content; 2–4% micro cellulose enhances structural integrity, while higher concentrations lead to a reduction in flexibility, rendering samples prone to fracture upon bending.

Electrical resistance measurements (Figure 7) reveal an exponential decrease in resistance between 10 and 20% PEDOT:PSS content, indicative of a percolation threshold within the PVA network. A 30% PEDOT:PSS loading further reduces electrical resistance, albeit to a lesser extent than the preceding increment. This behavior is representative of a conductive composite, with the point of significant resistance drop often referred to as the percolation threshold. At this threshold, conductive domains become sufficiently numerous and proximate to establish conductive pathways for electron transport. Further increases in PEDOT:PSS concentration yield only minor reductions in resistance, as the system becomes saturated once an adequate number of electron pathways have been established [55,56]. The results presented in Figure 7B suggest the presence of a percolation threshold between 10% and 20% PEDOT:PSS content within the PVA network. An additional saturation effect is observed with the incorporation of micro cellulose; while the addition of micro cellulose up to a content of 4% supports the formation of conductive paths, further addition hinders the formation of such paths due to the introduction of excessive non-conductive material into the composite. These findings suggest an optimal filling volume of 4% for micro cellulose, with a 20% PEDOT:PSS content being sufficient for the formation of conductive pathways.

Despite SEM examination suggesting the presence of pores between individual micro cellulose strands, porosity was not measurable with the pore size meter, indicating that the gaps between the micro cellulose fibers are filled with PVA or PEDOT:PSS, effectively sealing them. Consistent sulfur levels observed in the EDS analysis can be attributed to matrix effects that reduce the detection efficiency of sulfur X-rays, especially when a dominant PVA matrix is present. Since the EDS technique is surface-sensitive, if PEDOT:PSS is embedded deeper within the PVA matrix and the surface is primarily composed of PVA, carbon and oxygen will be predominantly detected. Additionally, the emitted X-rays from sulfur can be attenuated or overlapped by the sample’s density or thickness, leading to inefficient detection. Furthermore, processes during sample preparation, such as drying, cross-linking, or phase separation, may result in an uneven distribution of PEDOT:PSS, contributing to inconsistent sulfur detection despite an increased PEDOT:PSS content. PVA/micro cellulose/PEDOT:PSS networks are a promising material for various applications such as paper electrodes, flexible energy storage devices, and bioengineering sensors.

The PU/PEDOT:PSS sponges exhibit the second-greatest thickness among the produced samples, attributed to the presence of sugar particles in the dispersion, which prevents the application of thin layers. As an illustrative example, PUP1S2 possesses an average layer thickness of 1.3 mm. Electrical resistance remains relatively consistent across all samples, irrespective of sugar content, and lies at around 63 kΩ. This consistent electrical resistance suggests that saturation has been achieved, contrary to the anticipated trend of increased resistance with higher sugar amounts, which would imply a reduced PU/PEDT:PSS content in the overall mixture. This finding can be rationalized by assuming the presence of sufficient conductive material in the mixture, regardless of sugar concentrations. Moreover, measuring the electrical resistance presents difficulties due to the large pores and fine measuring tips, leading to inconsistent contact and unreliable readings.

The PUP1S1-2.5 samples display the largest measured pores among all tested samples. These sizable pores likely account for the notably lower presence of sulfur observed in the EDS analysis (Figure 11) compared to all other samples. The enhanced porosity allows X-rays to penetrate the sample more deeply, potentially resulting in a lower signal for lighter elements such as sulfur. Moreover, it can be hypothesized that the combination of polyurethane (PU) and PEDOT:PSS results in the encapsulation of PEDOT:PSS within the PU matrix. This encapsulation phenomenon may create a physical barrier, limiting the accessibility of X-rays during EDS analysis and hindering effective interaction with elements present in PEDOT:PSS, particularly sulfur. The combination of increased porosity, the inherent limitations of EDS in detecting light elements at higher depths, and the potential encapsulation of PEDOT:PSS in PU may all contribute to the challenge of accurately quantifying sulfur in this particular sample.

The thick and flexible PU/PEDOT:PSS sponges are ideal for use in pressure and strain sensors. Their versatility makes them well-suited for integration into wearable devices that monitor physiological parameters like pulse, respiration, and body movements. These applications are particularly relevant for fitness tracking, rehabilitation, and healthcare monitoring.

The production of puffed PEDOT:PSS structures involves a straightforward process of mixing a commercially available puffy printing paste with varying amounts of doped PEDOT:PSS, followed by drying in the oven for several minutes. The quantity of PEDOT:PSS also affects the puffing ability and mechanical stability. Samples with no PEDOT:PSS (TS1P0) exhibit a highly puffy structure that allows for light stretching, while those containing 400% PEDOT:PSS (TS1P4) are thinner and more prone to tearing. In that way, TS1P4 exhibits a mean layer thickness of 301.5 µm. Microscopic images illustrate that samples with increased PEDOT:PSS content lose their puffiness, causing the microbeads in the puffy printing paste to form a less interconnected surface. This change leads to a loss of structural integrity and an increase in pore size.

A PEDOT:PSS content of 200% (TS1P2) is necessary for adequate conductivity. The most significant decrease in electrical resistance occurs between samples TS1P1 (57.6 kΩ) and TS1P2 (3.66 kΩ), indicating a percolation threshold between 100% and 200% PEDOT:PSS content relative to Tubiscreen EX-TS-FF. Beyond this point, further addition only marginally reduces resistance, suggesting saturation.

Due to their soft and porous nature, puffed PEDOT:PSS structures are suitable to be integrated into biomedical sensors for monitoring various physiological parameters. They can be used in applications such as skin patches for continuous health monitoring and wound healing sensors.

The thickness of starch-based puffed PEDOT:PSS structures can be controlled by adjusting the amount of baking soda in the mixture. Sample F1P1.5BS3, for instance, has the highest layer thickness of all produced samples, which is not even measurable using the applied method. Moreover, pore size increases with higher baking soda content, though not reaching the pore size of PUP1S1-2.5. The presence of sizable pores is also the reason for the low presence of sulfur during EDS mapping.

While the mechanical properties are generally satisfactory, the samples can be easily torn apart. One main drawback is the samples’ susceptibility to water, caused by their natural character: when in contact with water, the samples soften and become gooey. The samples exhibit an electrical resistance of 426–441 kΩ, which is comparatively high in relation to the other samples.

The unique puffed structure, cost-effectiveness, and biocompatibility of these materials make them ideal for pressure and strain sensor applications. Their suitability arises from their ability to deform under pressure or strain, resulting in measurable changes in electrical resistance, which makes them effective for detecting mechanical forces.

Lastly, porous non-wovens are subjected to a spraying process with doped PEDOT:PSS (DMSO + glycerol), employing an increasing number of spraying layers (2, 4, 8, 16). Upon drying, samples sprayed with 8–16 layers exhibit a softer texture compared to those with 2–4 layers. NWP16, in particular, feels as if not completely dried through, attributed to the glycerol and its water-binding properties. Independently of the number of spraying layers, all samples of NWP2-16 remain thin. Microscopic examination reveals the gluing together of individual filaments with an increasing number of spraying layers. However, porosity measurements indicate the sustained presence of pores, albeit decreasing in size. Notably, NWP16 exhibits the smallest mean pore size of all measured porous structures (excluding PVA/micro cellulose/PEDOT:PSS structures) at 16 µm.

Electrical resistance measurements on sprayed non-wovens yield the lowest resistance values in the Ω-range for the first time. While 2 layers still exhibit relatively high electrical resistance (38.15 kΩ), spraying with 16 layers reduces the electrical resistance to 31.34 Ω. The low electrical resistance may be attributed to the superficial surface wetting of the samples, in contrast to the complete networking seen in other manufacturing methods. This allows for the better formation of conductive pathways, as the material does not need to be fully penetrated.

Sprayed non-wovens offer excellent flexibility and conductivity, making them ideal for flexible electronic devices. These materials can serve as electrodes in flexible displays, touch screens, wearable electronics, and various sensors.

## 5. Conclusions

This study introduces novel and facile methods for the fabrication of diverse porous PEDOT structures and provides a comprehensive characterization of their porosity, electrical resistance, and morphology. The incorporation of micro cellulose, the utilization of blowing agents or scaffolding materials, and the implementation of spraying techniques enable the fabrication of a wide array of porous structures, spanning from thin and slightly porous to thick and highly porous.

The PU/PEDOT sponges and starch-based puffed samples represent thicker porous structures, exhibiting a distinct 3-dimensional character and large pores. These structures demonstrate relatively high electrical resistance and face limitations in terms of mechanical stability and water resistance. In contrast, thinner PEDOT:PSS structures (P10-30C0-16, TS1P1-4, NWP2-16) possess smaller pores and achieve lower electrical resistance, which can be modulated through the PEDOT:PSS content within the composite. These findings highlight the potential of these methods for creating tailored porous conductive polymer structures suitable for applications in sensors, hydrogels, and supercapacitors.

Future research endeavors must focus on the integration of these versatile materials into flexible wearables and smart textiles, as well as exploring further applications. The successful incorporation of these porous PEDOT:PSS structures into such systems would unlock a myriad of possibilities in the realm of wearable electronics, enabling the development of highly sensitive and responsive devices. Moreover, the unique combination of porosity, conductivity, and tunability exhibited by these structures opens up avenues for their utilization in advanced energy storage systems, such as supercapacitors, where high surface area and efficient charge transport are paramount.

To fully realize the potential of these porous PEDOT:PSS structures, it is imperative to conduct further studies aimed at optimizing their properties and addressing any limitations. This may involve investigating alternative blowing agents, scaffolding materials, or processing techniques to enhance mechanical stability and water resistance, particularly in the case of PU/PEDOT sponges and starch-based puffed structures. Additionally, exploring the effects of different dopants or post-treatment methods on the electrical and morphological properties of these structures could provide valuable insights into fine-tuning their performance for specific applications.

In conclusion, this study lays the groundwork for the development of a new class of porous conductive polymer structures with immense potential in various fields. The presented fabrication methods, coupled with the comprehensive characterization of the resulting structures, provide a solid foundation for future research and development efforts. As the demand for advanced materials in wearable electronics, smart textiles, and energy storage continues to grow, the porous PEDOT:PSS structures introduced in this study are poised to play a significant role in shaping the future of these technologies.

## Figures and Tables

**Figure 1 sensors-24-04919-f001:**
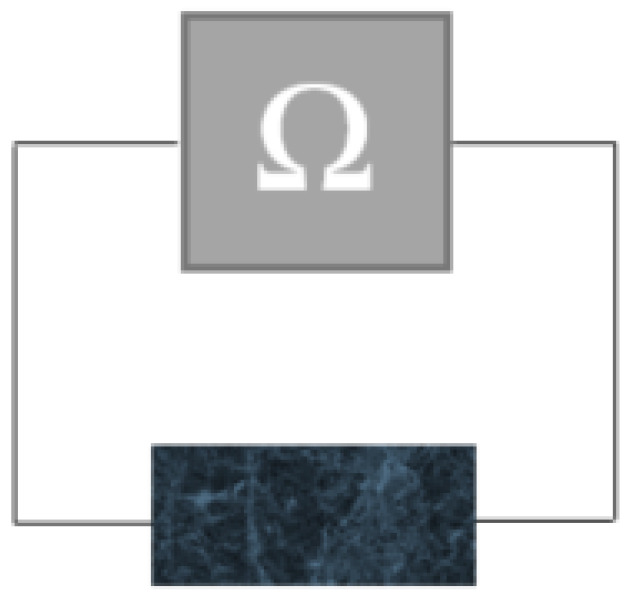
Resistance measurement set-up.

**Figure 2 sensors-24-04919-f002:**
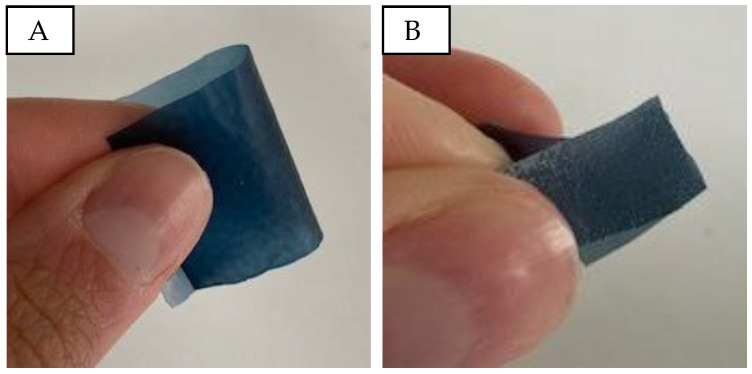
Bending and touch of PVA/micro cellulose/PEDOT:PSS samples. (**A**): P20C2; (**B**): P20C16.

**Figure 3 sensors-24-04919-f003:**
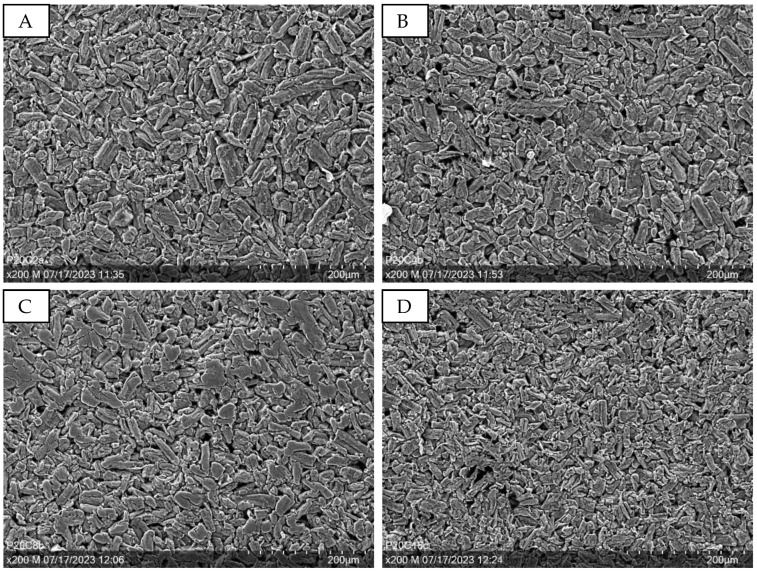
SEM images at 200× magnification showing the surface of the PVA/micro cellulose/ PEDOT:PSS networks with 20% PEDOT:PSS and different micro cellulose contents. (**A**): P20C2; (**B**): P20C4; (**C**): P20C8; (**D**): P20C16.

**Figure 4 sensors-24-04919-f004:**
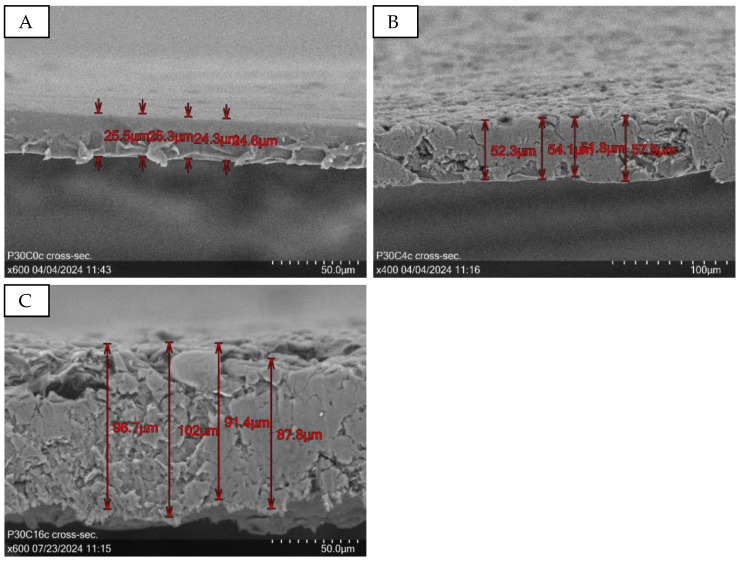
Cross-section of P30C0a (**A**) at 600× magnification with a mean layer thickness of 25.9 µm, P30C4c (**B**) at 400× magnification with a mean layer thickness of 53.9 µm, and P30C16c (**C**) with a mean layer thickness of 94.5 µm.

**Figure 5 sensors-24-04919-f005:**
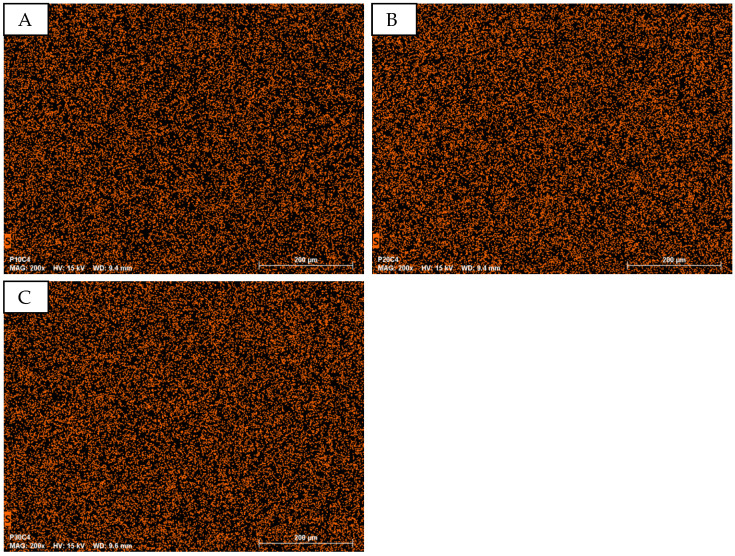
EDS mapping of the element sulfur at different PEDOT:PSS but same micro cellulose content in different PVA/micro cellulose/PEDOT:PSS samples. (**A**): P10C4; (**B**): P20C4; (**C**): P30C4.

**Figure 6 sensors-24-04919-f006:**
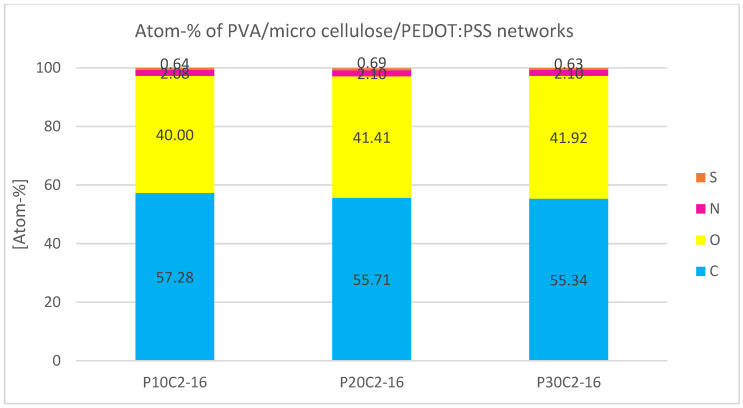
Atomic percentage of the elements found during EDS analysis of PVA/micro cellulose/PEDOT:PSS networks.

**Figure 7 sensors-24-04919-f007:**
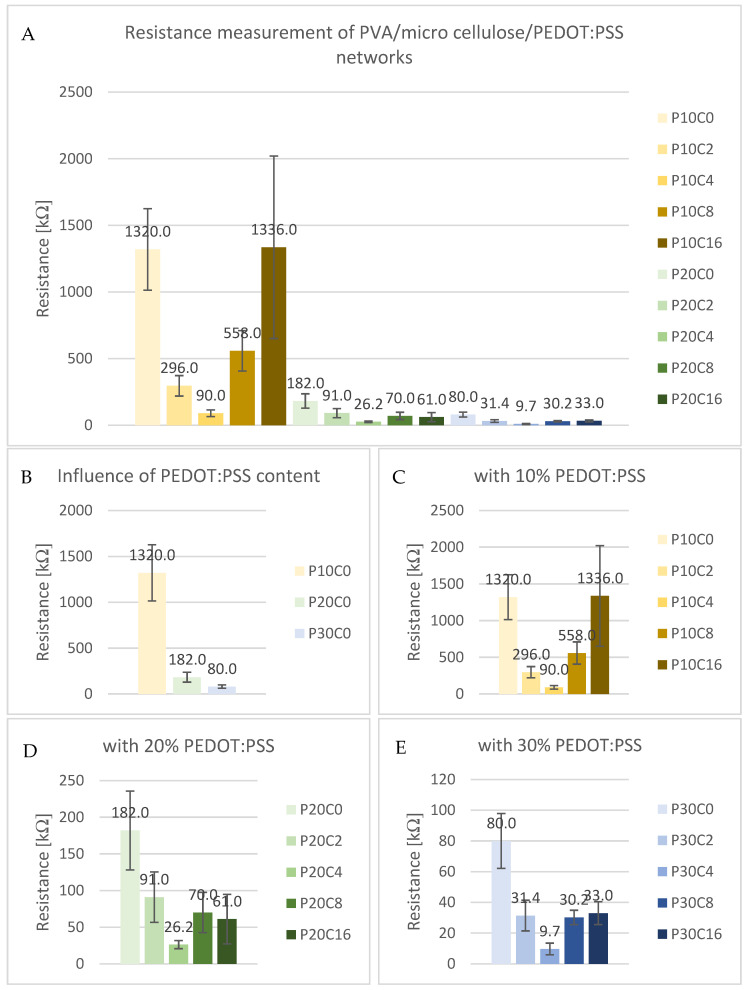
Resistance measurement of PVA/micro cellulose/PEDOT:PSS network shows not only a decrease in resistance with increasing PEDOT:PSS content but also due to the addition of micro cellulose up to a certain degree and the saturation of the samples. (**A**): Overview of all samples; (**B**): Influence of PEDOT:PSS content; (**C**): Samples with 10% PEDOT:PSS and different micro cellulose contents; (**D**): Samples with 20% PEDOT:PSS and different micro cellulose contents; (**E**): Samples with 30% PEDOT:PSS and different micro cellulose contents.

**Figure 8 sensors-24-04919-f008:**
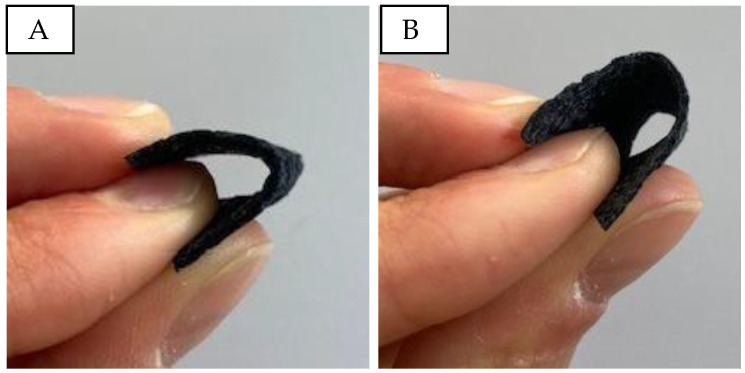
Bending and touch of porous PU/PEDOT:PSS sponges. (**A**): PUP1S2; (**B**): PUP1S2.5.

**Figure 9 sensors-24-04919-f009:**
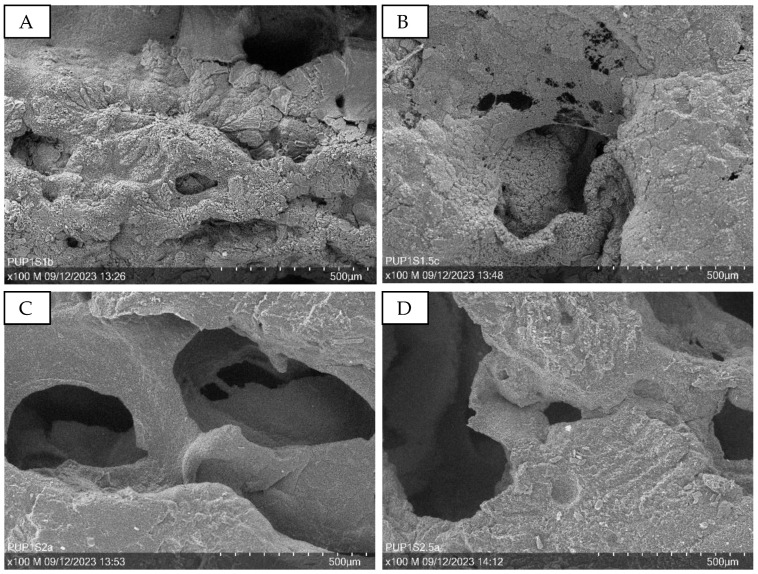
SEM images at 100× magnification showing the surface of the porous PU/PEDOT:PSS sponges produced with different amounts of sugar. (**A**): PUP1S1; (**B**): PUP1S1.5; (**C**): PUP1S2; (**D**): PUP1S2.5.

**Figure 10 sensors-24-04919-f010:**
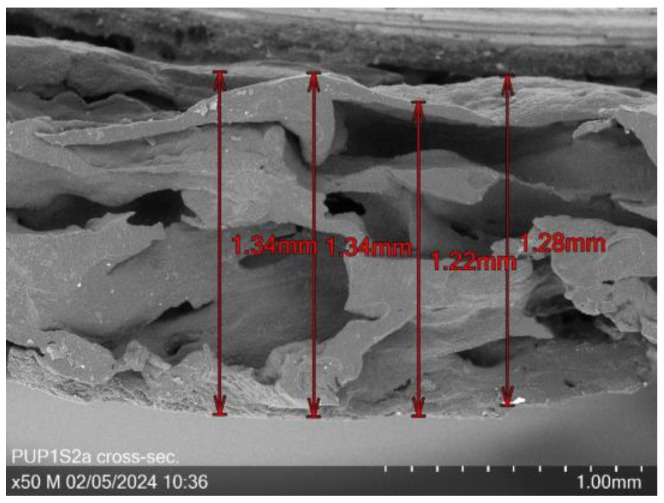
Cross-section of PUP1S2a at 50× magnification with a mean layer thickness of 1.3 mm.

**Figure 11 sensors-24-04919-f011:**
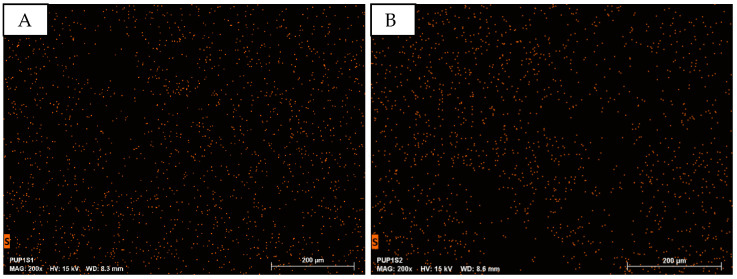
EDS mapping of the element sulfur in the samples of PU/PEDOT:PSS sponges prepared with different sugar concentrations. (**A**): PUP1S1; (**B**): PUP1S2.

**Figure 12 sensors-24-04919-f012:**
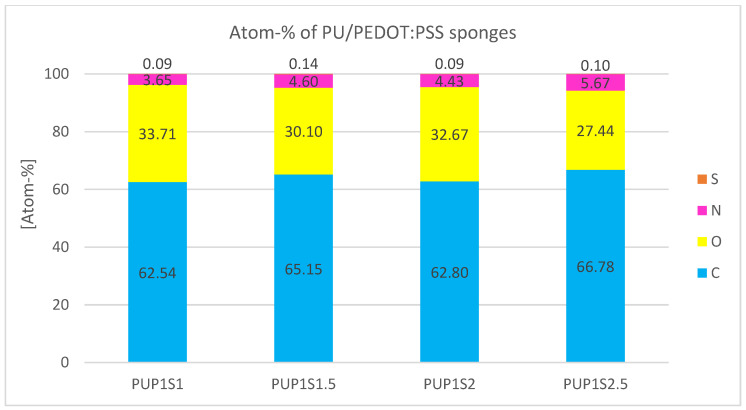
Atomic percentage of the elements found during EDS analysis of PU/PEDOT:PSS sponges.

**Figure 13 sensors-24-04919-f013:**
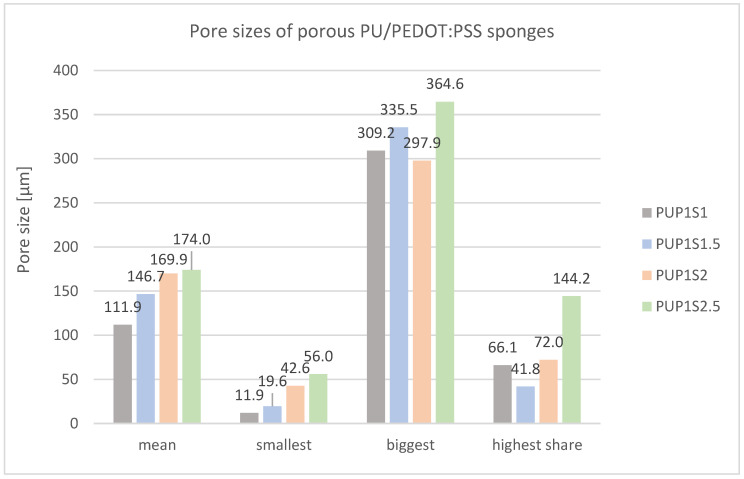
Mean pore size and pore size distribution of porous PU/PEDOT:PSS sponges produced with different sugar contents.

**Figure 14 sensors-24-04919-f014:**
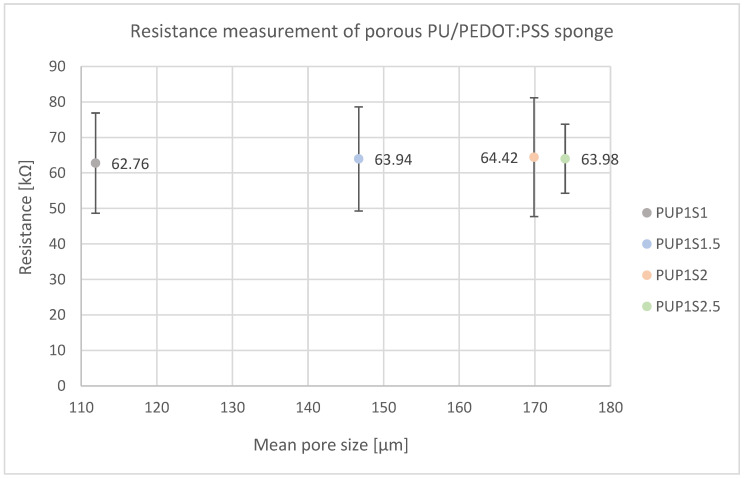
Resistance measurement of porous PU/PEDOT:PSS sponges shows no difference in electrical resistance with different sugar contents and thus resulting pore sizes.

**Figure 15 sensors-24-04919-f015:**
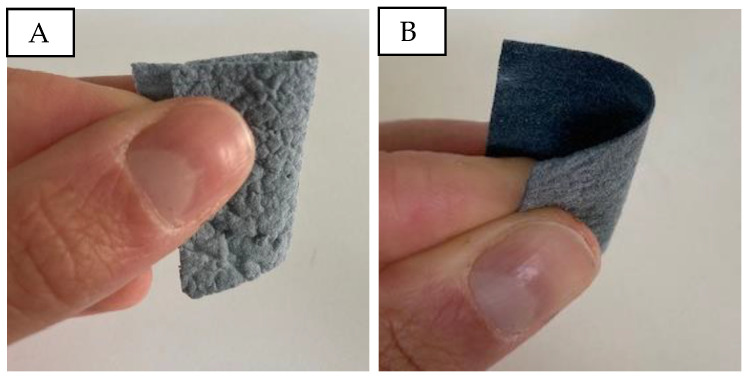
Bending and touch of the puffed PEDOT:PSS structure. (**A**): TS1P1; (**B**): TS1P4.

**Figure 16 sensors-24-04919-f016:**
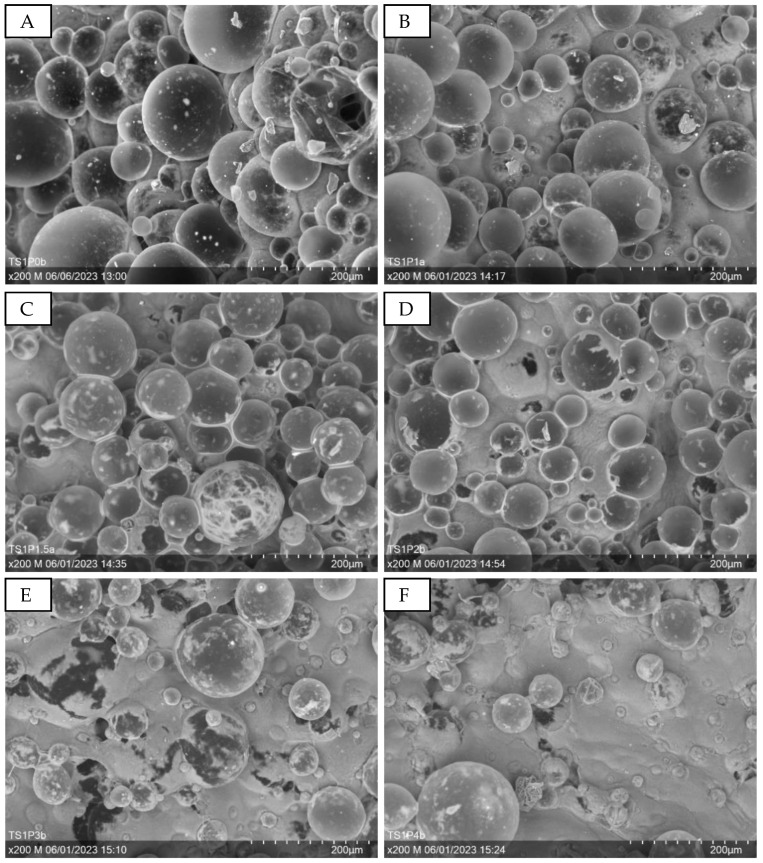
SEM images at 200× magnification showing the surface of the puffed PEDOT:PSS structure at different PEDOT:PSS contents. (**A**): 0% PEDOT:PSS—TS1P0; (**B**): 100% PEDOT:PSS—TS1P1; (**C**): 150% PEDOT:PSS—TS1P1.5; (**D**): 200% PEDOT:PSS—TS1P2; (**E**): 300% PEDOT:PSS—TS1P3; (**F**): 400% PEDOT:PSS—TS1P4.

**Figure 17 sensors-24-04919-f017:**
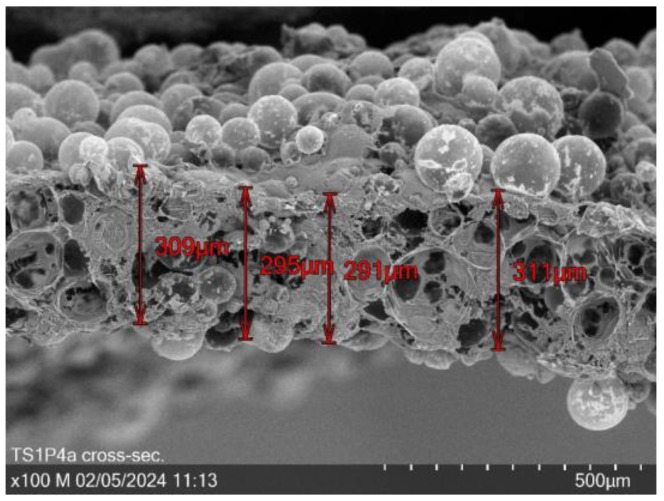
Cross-section of TS1P4b at 100× magnification with a mean layer thickness of 301.5 µm.

**Figure 18 sensors-24-04919-f018:**
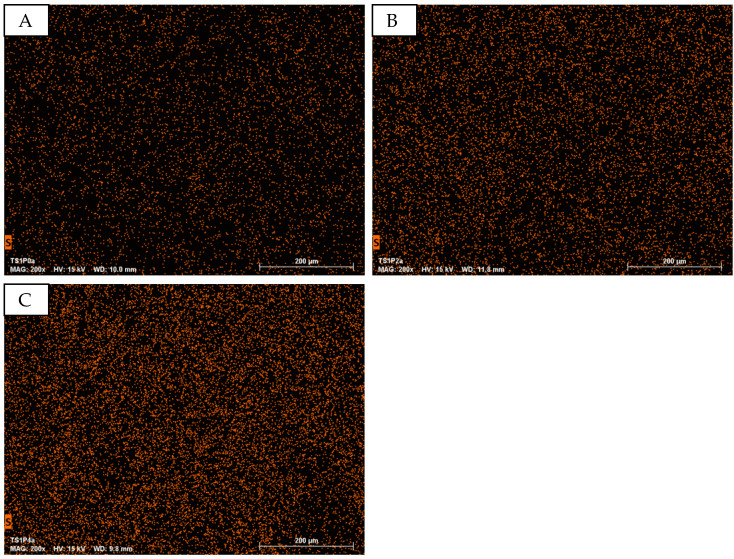
EDS mapping of the element sulfur at different PEDOT:PSS concentrations in the puffed PEDOT:PSS structures. (**A**): TS1P0; (**B**): TS1P2; (**C**): TS1P4.

**Figure 19 sensors-24-04919-f019:**
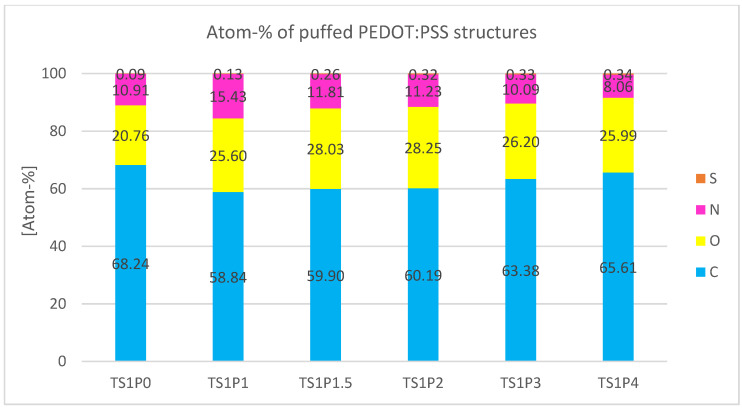
Atomic percentage of the elements found during EDS analysis of porous puffed PEDOT:PSS samples.

**Figure 20 sensors-24-04919-f020:**
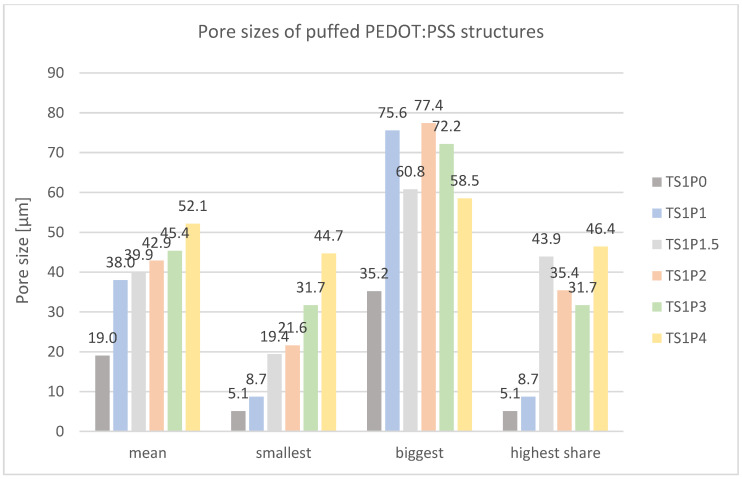
Mean pore size and pore size distribution of puffed PEDOT:PSS structures with different PEDOT:PSS contents.

**Figure 21 sensors-24-04919-f021:**
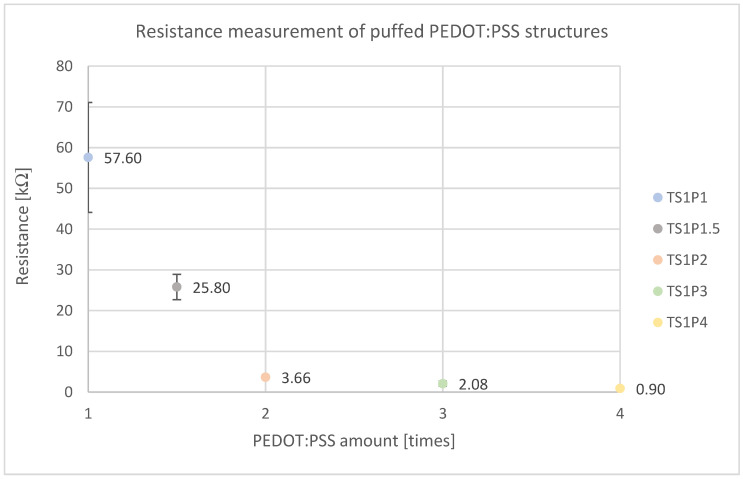
Resistance measurement of puffed PEDOT:PSS structure showing a decrease in resistance with increasing amount of PEDOT:PSS.

**Figure 22 sensors-24-04919-f022:**
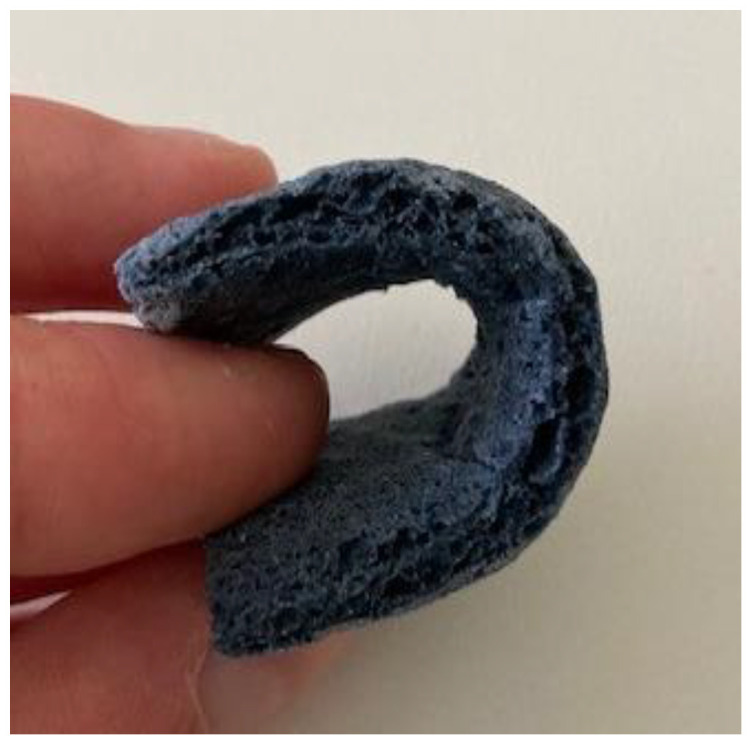
Bending and touch of starch based puffed PEDOT:PSS structures.

**Figure 23 sensors-24-04919-f023:**
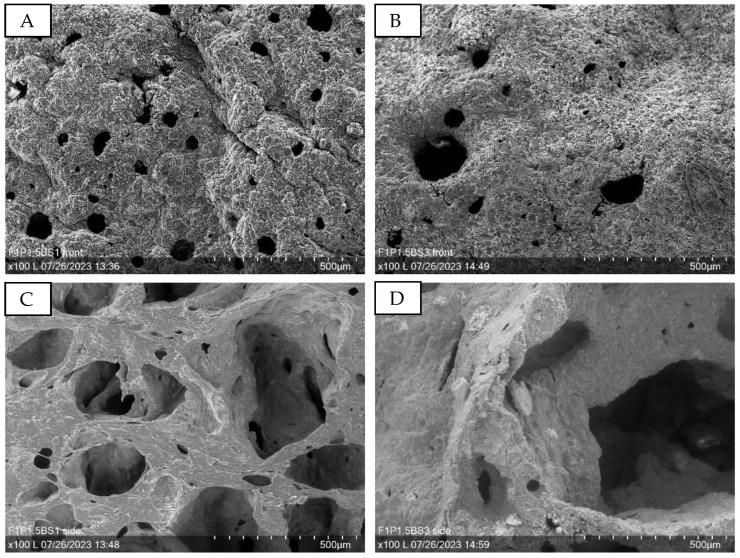
SEM examination (100×) of puffed PEDOT:PSS samples produced with starch based puffy paint. (**A**): F1P1.5BS1—top view; (**B**): F1P1.5BS3—top view; (**C**): F1P1.5BS1—cross-sectional view; (**D**): F1P1.5B3—cross-sectional view.

**Figure 24 sensors-24-04919-f024:**
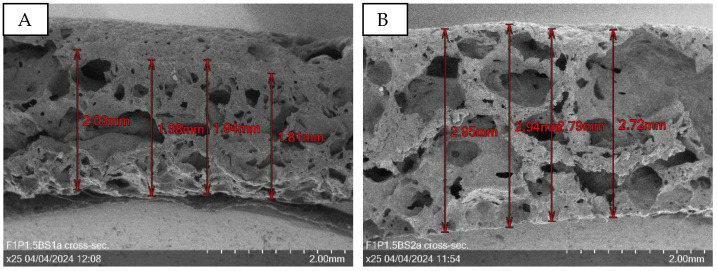
Cross-section of F1P1.5BS1a at 25× magnification with a mean layer thickness of 1.94 mm (**A**) and F1P1.5BS2a at 25× magnification with a mean layer thickness of 2.85 mm (**B**).

**Figure 25 sensors-24-04919-f025:**
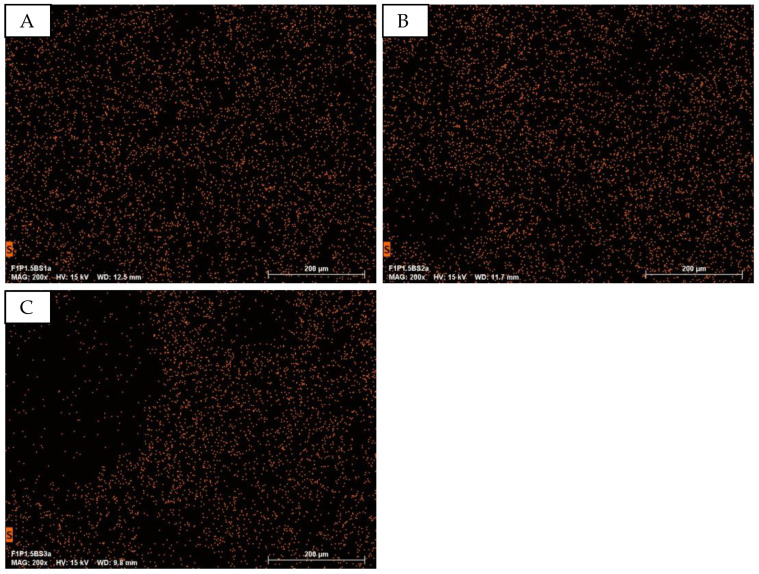
EDS mapping of the element sulfur at different baking soda concentrations in the starch-based puffed PEDOT:PSS structures. (**A**): F1P1.5BS1; (**B**): F1P1.5BS2; (**C**): F1P1.5BS3.

**Figure 26 sensors-24-04919-f026:**
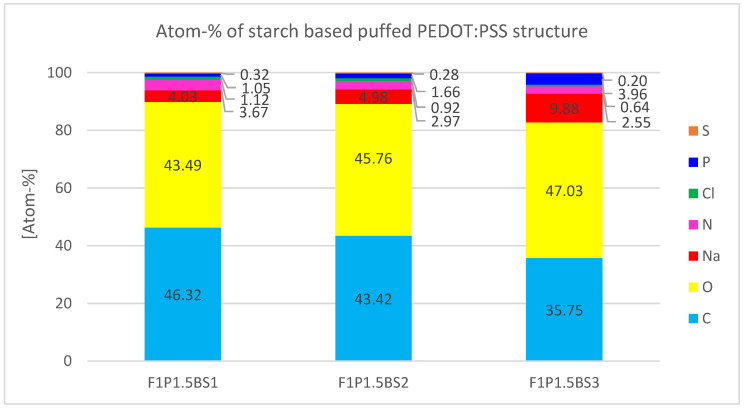
Atomic percentage of the elements found during EDS analysis of porous starch-based puffed PEDOT:PSS samples.

**Figure 27 sensors-24-04919-f027:**
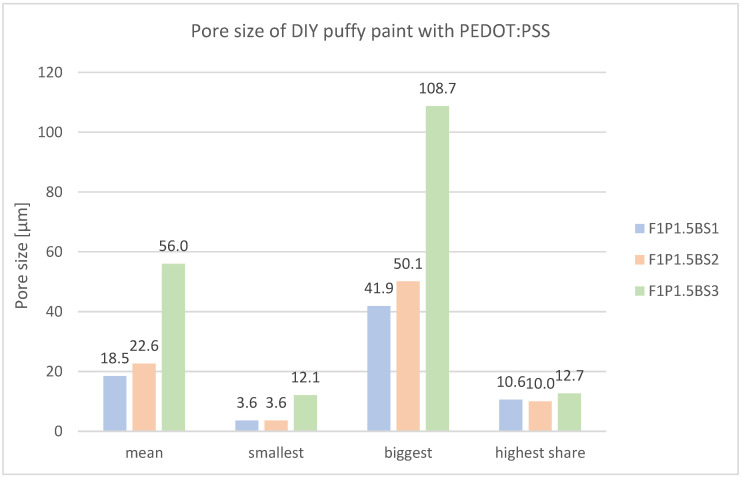
Mean pore size and pore size distribution of starch-based puffed PEDOT:PSS samples with different amounts of baking soda.

**Figure 28 sensors-24-04919-f028:**
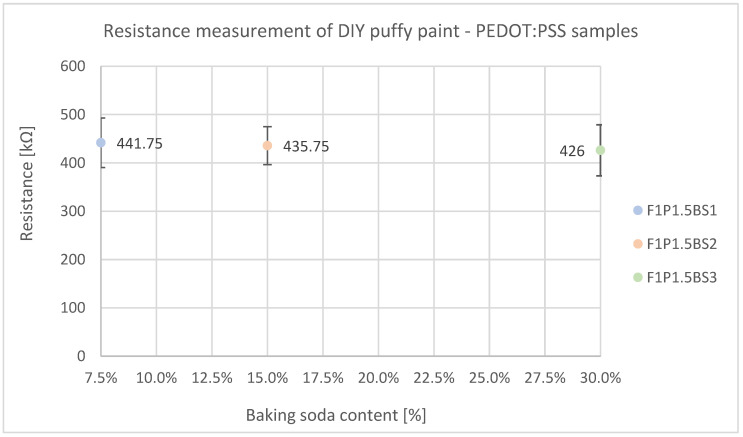
Electrical resistance measurement of starch-based puffed PEDOT:PSS samples show no significant change in electrical resistance in dependence on baking soda content.

**Figure 29 sensors-24-04919-f029:**
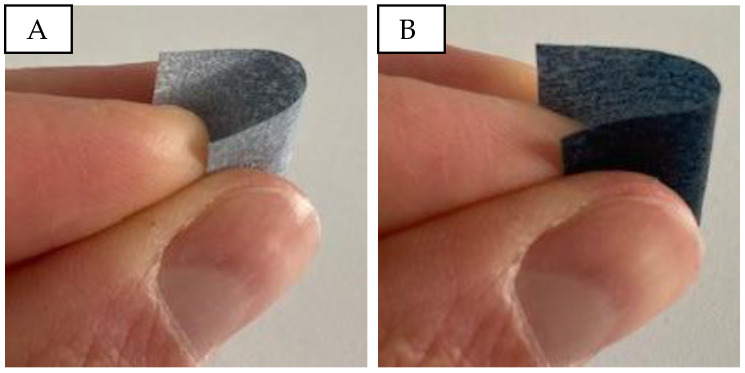
Bending and touch of PEDOT:PSS sprayed non-wovens. (**A**): NWP2; (**B**): NWP16.

**Figure 30 sensors-24-04919-f030:**
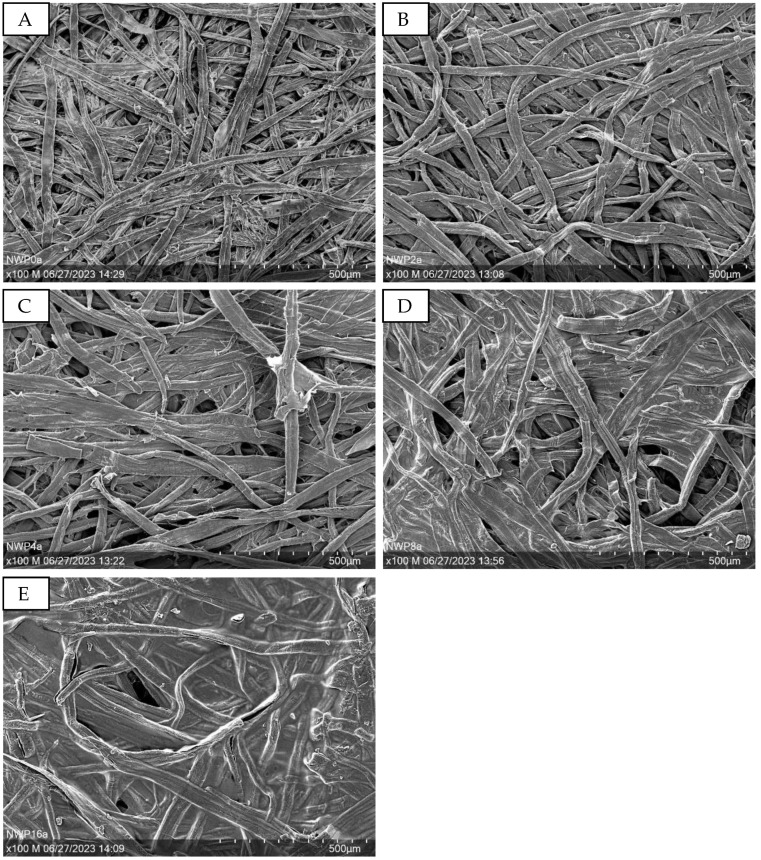
SEM images at 100x magnification of the PEDOT:PSS sprayed non-wovens with different layers of PEDOT:PSS. (**A**): NWP0, 0 layers; (**B**): NWP2, 2 layers; (**C**): NWP4, 4 layers; (**D**): NWP8, 8 layers; (**E**): NWP16, 16 layers.

**Figure 31 sensors-24-04919-f031:**
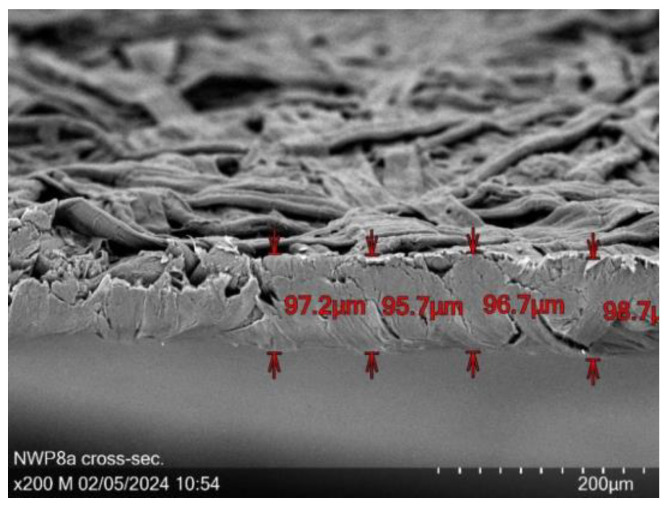
Cross-section of NWP8a at 200× magnification with a mean layer thickness of 97.1 µm.

**Figure 32 sensors-24-04919-f032:**
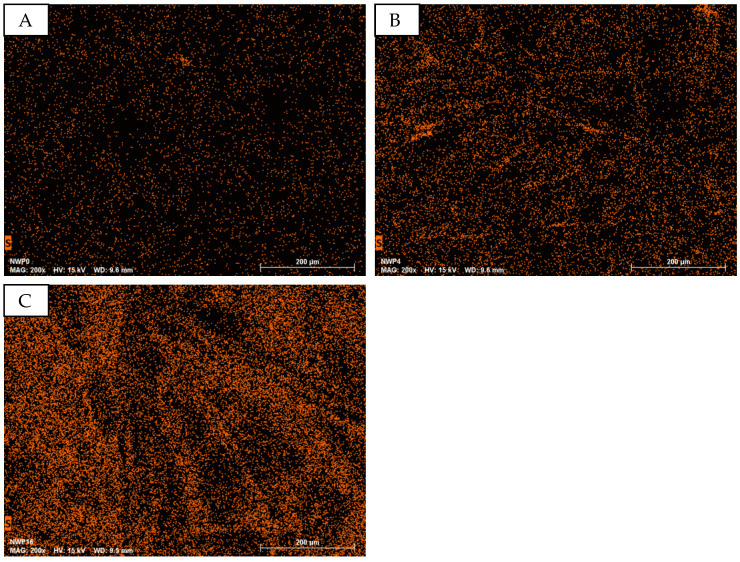
EDS mapping of the element sulfur in PEDOT:PSS sprayed non-woven samples with an increasing number of spraying cycles. (**A**): NWP0; (**B**): NWP4; (**C**): NWP16.

**Figure 33 sensors-24-04919-f033:**
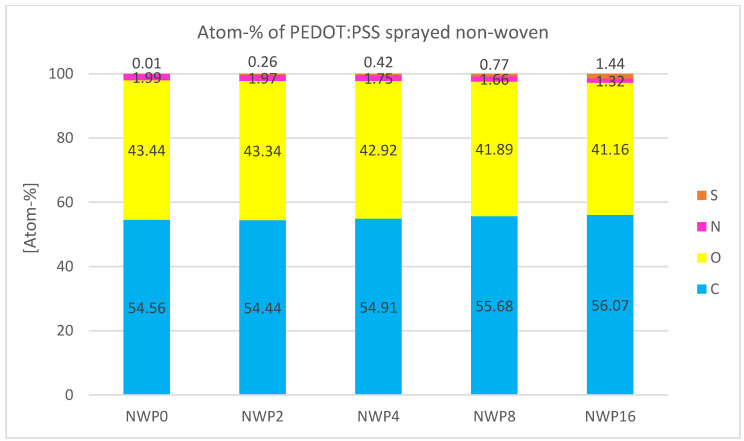
Atomic percentage of the elements found during EDS analysis of PEDOT:PSS-sprayed non-woven samples.

**Figure 34 sensors-24-04919-f034:**
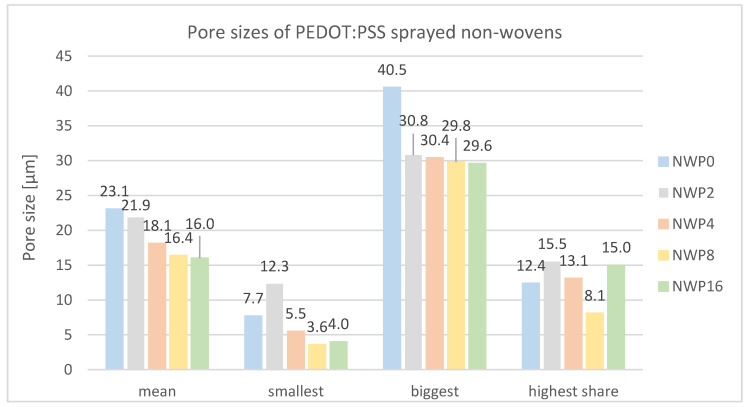
Mean pore size and pore size distribution of PEDOT:PSS-sprayed non-wovens produced by different spraying cycles.

**Figure 35 sensors-24-04919-f035:**
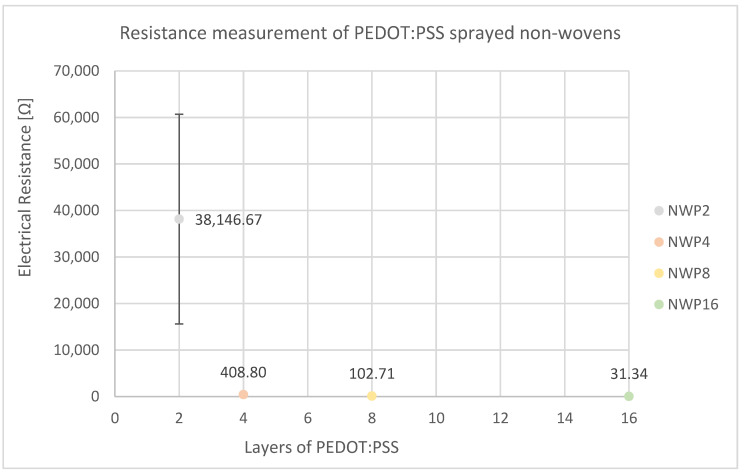
Resistance measurement of PEDOT:PSS sprayed non-wovens shows that resistance decreases with the number of sprayed layers.

**Table 1 sensors-24-04919-t001:** Mixing ratio of PVA/micro cellulose/PEDOT:PSS networks.

PEDOT:PSS Content (wt%)	Micro Cellulose Content (wt%)	Designation
10	0	P10C0
	2	P10C2
	4	P10C4
	8	P10C8
	16	P10C16
20	0	P20C0
	2	P20C2
	4	P20C4
	8	P20C8
	16	P20C16
30	0	P30C0
	2	P30C2
	4	P30C4
	8	P30C8
	16	P30C16

**Table 2 sensors-24-04919-t002:** Mixing ratio and designation of PU/PEDOT:PSS sponges.

Ratio PU/PEDOT:PSS and Sugar	Designation
1:1	PUP1S1
1:1.5	PUP1S1.5
1:2	PUP1S2
1:2.5	PUP1S2.5

**Table 3 sensors-24-04919-t003:** Mixing ratio and designation of puffed PEDOT:PSS structures.

Ratio Tubiscreen and PEDOT:PSS	Designation
1:0	TS1P0
1:1	TS1P1
1:1.5	TS1P1.5
1:2	TS1P2
1:3	TS1P3
1:4	TS1P4

**Table 4 sensors-24-04919-t004:** Mixing ratio and designation of starch-based puffed PEDOT:PSS structures.

Baking Soda Amount (wt%)	Designation
7.5	F1P1.5BS1
15	F1P1.5BS2
30	F1P1.5BS3

**Table 5 sensors-24-04919-t005:** Number of spraying layers and designation of PEDOT:PSS sprayed non-wovens.

No. of Spraying Layers	Designation
0	NWP0
2	NWP2
4	NWP4
8	NWP8
16	NWP16

**Table 6 sensors-24-04919-t006:** Layer thickness of all PVA/micro cellulose/PEDOT:PSS networks with 30% PEDOT:PSS.

Designation	Layer Thickness [µm]
P30C0	25.9
P30C2	41.0
P30C4	53.9
P30C8	81.8
P30C16	94.5

## Data Availability

The data presented in this study are available on request from the corresponding author.

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
