# Peer review of "Straightforward Production Methods for Diverse Porous PEDOT:PSS Structures and Their Characterization"

_sensors, 2024, doi:10.3390/s24154919_

Round 1

Reviewer 1 Report

Comments and Suggestions for Authors

In this manuscript, the authors discussed various processes for the preparation of porous conducting polymer PEDOT materials, followed by a detailed description of their pore densities, conductivities, and morphological characteristics. The foaming techniques included in the article are relatively comprehensive, the research is in general well conducted. Thus I recommend publication of this work after the authors have addressed the following issues:

1.    In the introduction, the authors focus on the effect of conductivity on porous conductive polymers but do not discuss the pore structure or the latest progress in pore space, pore density, and pore-making technology from previous works. The introduction should provide a general overview of the entire text and highlight the research focus.

2.    In the second part, the characterization and discussion of each manufacturing method only discuss the difference between the added and unadded microcellulose layer thickness. The effect of different component ratios on the average layer thickness should be further explored, particularly regarding the non-linear effect of varying levels of microcellulose addition.

3.    There are minor issues with the picture labeling. For instance, in Figure 7, it is difficult to visualize the specific resistance value of each component, making it challenging to extract desired data from similar data points. Additionally, the numerical labeling in Figure 33 is inconsistent, with some labels inside the columns and some outside. The figures should adhere to standardized specifications.

4.    The discussion on the average pore size and distribution can be better represented using a normal distribution diagram. This would provide a more intuitive understanding of the average pore size.

5.    The article should systematically compare the pore structure and conductivity of the five methods. This comparison should be presented in the form of images, allowing readers to quickly identify which manufacturing method suits their needs.

6.    The article mentions that the prepared materials can be applied in sensing and other functional equipment. The authors should select the optimal structure after a systematic comparison and supplement the article with relevant sensing data. This would validate the authors' claims and support the practical application of the materials.

Comments on the Quality of English Language

good

Author Response

In this manuscript, the authors discussed various processes for the preparation of porous conducting polymer PEDOT materials, followed by a detailed description of their pore densities, conductivities, and morphological characteristics. The foaming techniques included in the article are relatively comprehensive, the research is in general well conducted. Thus I recommend publication of this work after the authors have addressed the following issues:

Thank you for your valuable feedback and your general recommendation for the publication of this study. We appreciate you taking the time to read the publication thoroughly!
We made some changes to the manuscript and hope that we could address the remarks that you pointed out, so that you like the article better now.

  1. In the introduction, the authors focus on the effect of conductivity on porous conductive polymers but do not discuss the pore structure or the latest progress in pore space, pore density, and pore-making technology from previous works. The introduction should provide a general overview of the entire text and highlight the research focus.

We revised the introduction and added a section on the use of porous polymers in sensing applications. This addition aims to better highlight the importance and application of porous conducting substrates.

  1. In the second part, the characterization and discussion of each manufacturing method only discuss the difference between the added and unadded microcellulose layer thickness. The effect of different component ratios on the average layer thickness should be further explored, particularly regarding the non-linear effect of varying levels of microcellulose addition.

      We have added a new micrograph featuring a higher concentration of microcellulose, which illustrates how the layer thickness increases with greater microcellulose concentrations. Additionally, we have included the layer thickness measurements for all P30C0-16 samples in the main text.

  1. There are minor issues with the picture labeling. For instance, in Figure 7, it is difficult to visualize the specific resistance value of each component, making it challenging to extract desired data from similar data points. Additionally, the numerical labeling in Figure 33 is inconsistent, with some labels inside the columns and some outside. The figures should adhere to standardized specifications.

      Thank you for that remark. We have improved these points.

  1. The discussion on the average pore size and distribution can be better represented using a normal distribution diagram. This would provide a more intuitive understanding of the average pore size.

      Unfortunately, the used measuring method and equipment does not display a normal distribution diagram. But we will consider it for next experiments.

  1. The article should systematically compare the pore structure and conductivity of the five methods. This comparison should be presented in the form of images, allowing readers to quickly identify which manufacturing method suits their needs.

       A direct comparison of these methods is challenging due to the inherent differences in the samples. Their properties can be controlled to a certain extent and should be adapted depending on the application. Therefore, a systematic comparison is difficult. However, we hope that we highlighted the differences in the discussion, where we also outline possible sensing applications.

  1. The article mentions that the prepared materials can be applied in sensing and other functional equipment. The authors should select the optimal structure after a systematic comparison and supplement the article with relevant sensing data. This would validate the authors' claims and support the practical application of the materials.

We have revised the discussion to include suggestions on potential sensor applications for these porous structures, aiming to clarify their usability. Unfortunately, application-oriented sensing data has not yet been obtained.

Reviewer 2 Report

Comments and Suggestions for Authors

Brendgen et al. produced a variety of porous PEDOT:PSS structures using incorporation of microfibrillated cellulose, the use of foaming agents, the formation of sponge-like structures, and spraying onto porous substrates, and performed detailed characterisation. Overall, the experiments were full and informative, but there are still some issues that need to be improved.

Before that, I would like to explain my feelings as a reader reading the whole article, in fact, I don't particularly recommend this style of writing, that is, to state the conclusion of the experiment and wait until the last part of the article and then go to some detailed explanations, which will cause some trouble for the reader to read. And in the body of the various experiments as well as some parts of the text, it is directly stated what experiments were done and what the results were, but there is no explanation as to why the experiments were done, what problems were overcome or some other reasons, which is less logical during this period and creates confusion in the reading. In addition, the four preparations in the article seem to be juxtaposed with each other, without some connection, which makes the content of the detailed basis of the various parts seem to feel a little fragmented without a sense of logic, and it will be tired to read. It is worth noting that the above is not a denial of the content of the article, but my personal opinions and feelings, do not need to make extensive changes. The following are some of the questions I have about the article, which must be answered or improved.

1. In order to identify again, whether the several preparations in the text are part of different preparation techniques that are not necessarily related to each other, or whether there is an inherent recursive relationship between them, e.g., the latter part of the text builds on the work done in the former part of the text. If there is a relationship, some additional clarification is needed to enhance the sense of logic, as this seems to me to be separate work. The introductory section spends a lot of time on PEDOT:PSS itself, but there needs to be more description of why the porous structure is needed, in order to distil the necessity and innovation of the work in this paper.

2. There is an overlap in the labelling of the figures in the Figure4 diagram, which is not readable.

3. The above paragraph of Figure4, what is the purpose here of looking at the cross-sectional area of the sample with or without the addition of cellulose? Was the same amount of solution used and made in a mould with the same base area? If it's to show that adding cellulose has a better network structure then there should be some representation of that in this section? I'm more curious if adding more amounts of cellulose (2%, 8%, 16%) the sample gets thicker or is it similar to Figure4B?

4. The EDS test in Figure5, which analyses samples with different levels of PEDOT:PSS, should show that sulphur is richer and denser? The levels of PEDOT:PSS are different, but the text says that the levels of sulphur are generally consistent, so why is that?

5. Figure14 Why do samples with different levels of sugar produce changes in porosity but not in electrical properties? The article didn't explain this until I read the discussion section at the end and saw that it said saturation could occur, but this explanation doesn't seem very convincing?

6. For the sake of rigour, some histograms may need error bars?

7. The text lists a variety of ways to fabricate porous PEDOT:PSS structures, what conclusions did you ultimately come to, and which method was better? What are the advantages and disadvantages of each? Or for what applications is each method suitable?

Comments on the Quality of English Language

None

Author Response

Brendgen et al. produced a variety of porous PEDOT:PSS structures using incorporation of microfibrillated cellulose, the use of foaming agents, the formation of sponge-like structures, and spraying onto porous substrates, and performed detailed characterisation. Overall, the experiments were full and informative, but there are still some issues that need to be improved.

Before that, I would like to explain my feelings as a reader reading the whole article, in fact, I don't particularly recommend this style of writing, that is, to state the conclusion of the experiment and wait until the last part of the article and then go to some detailed explanations, which will cause some trouble for the reader to read. And in the body of the various experiments as well as some parts of the text, it is directly stated what experiments were done and what the results were, but there is no explanation as to why the experiments were done, what problems were overcome or some other reasons, which is less logical during this period and creates confusion in the reading. In addition, the four preparations in the article seem to be juxtaposed with each other, without some connection, which makes the content of the detailed basis of the various parts seem to feel a little fragmented without a sense of logic, and it will be tired to read. It is worth noting that the above is not a denial of the content of the article, but my personal opinions and feelings, do not need to make extensive changes. The following are some of the questions I have about the article, which must be answered or improved.

Thank you for your valuable feedback and your general recommendation for the publication of this study. We appreciate you taking the time to read the publication thoroughly! We made some changes to the manuscript and hope that we could address the remarks that you pointed out, so that you like the article better now.

We will take your remarks about the structure into account for future publications. However, for this submission, the journal requires that the results and discussion be presented as separate sections.

  1. In order to identify again, whether the several preparations in the text are part of different preparation techniques that are not necessarily related to each other, or whether there is an inherent recursive relationship between them, e.g., the latter part of the text builds on the work done in the former part of the text. If there is a relationship, some additional clarification is needed to enhance the sense of logic, as this seems to me to be separate work. The introductory section spends a lot of time on PEDOT:PSS itself, but there needs to be more description of why the porous structure is needed, in order to distil the necessity and innovation of the work in this paper.

We revised the introduction and added a section on the use of porous polymers in sensing applications. This addition aims to better highlight the importance and application of porous conducting substrates.

  1. There is an overlap in the labelling of the figures in the Figure4 diagram, which is not readable.

Unfortunately, we cannot correct that overlap anymore as it comes like that from the SEM. However, it is 51.8 and 54.1µm.

  1. The above paragraph of Figure4, what is the purpose here of looking at the cross-sectional area of the sample with or without the addition of cellulose? Was the same amount of solution used and made in a mould with the same base area? If it's to show that adding cellulose has a better network structure then there should be some representation of that in this section? I'm more curious if adding more amounts of cellulose (2%, 8%, 16%) the sample gets thicker or is it similar to Figure4B?

We added another micrograph that shows that with even more micro cellulose layer thickness increases. Also, we have given all values for the samples P30C0-16 in a table, so you can see the increase in layer thickness.

  1. The EDS test in Figure5, which analyses samples with different levels of PEDOT:PSS, should show that sulphur is richer and denser? The levels of PEDOT:PSS are different, but the text says that the levels of sulphur are generally consistent, so why is that?

We added assumptions about that finding in the discussion and we assume it has something to do with matrix effects that reduce the detection efficiency of sulfur X-rays, especially when a dominant PVA matrix is present.

  1. Figure14 Why do samples with different levels of sugar produce changes in porosity but not in electrical properties? The article didn't explain this until I read the discussion section at the end and saw that it said saturation could occur, but this explanation doesn't seem very convincing?

Another possible explanation could be measurement errors arising from assessing the resistance of substrates with large pores with fine measurement tips. we elaborate on this effect in the discussion.

  1. For the sake of rigour, some histograms may need error bars?

We have added the standard deviation to the resistance measurement figures. Unfortunately, we cannot include the standard deviation in the pore size measurement figures because they also display absolute values.

  1. The text lists a variety of ways to fabricate porous PEDOT:PSS structures, what conclusions did you ultimately come to, and which method was better? What are the advantages and disadvantages of each? Or for what applications is each method suitable?

We have revised the discussion to include potential sensing applications for each method. Given the significant differences between the substrates, it's difficult to determine which is superior; it ultimately depends on the specific application.

Round 2

Reviewer 2 Report

Comments and Suggestions for Authors

No